# Review article: Snow and ice avalanches in high mountain Asia – scientific, local and indigenous knowledge

Anushilan Acharya[1], Jakob F. Steiner[2,3], Khwaja Momin Walizada[4], ~~Zakir Hussain Zakir[5],~~ Salar Ali[5], Zakir Hussain Zakir[5], -Arnaud Caiserman[6], Teiji Watanabe[7]

[1]Graduate School of Environmental Science, Hokkaido University, Sapporo 060-0810, Japan
[2]International Centre for Integrated Mountain Development, Lalitpur, Nepal
[3]University of Graz, Department of Geography and Regional Science, Heinrichstrasse 36, 8010 Graz, Austria
[4]Aga Khan Agency for Habitat, Kabul, Afghanistan
[5]Department of Environmental Science, University of Baltistan 16100, Skardu, Pakistan
[6]Mountain Societies Research Institute, University of Central Asia, Khorog, 736000, Tajikistan
[7]Faculty of Environmental Earth Science and Graduate School of Environmental Science, Hokkaido University, Japan

*Correspondence to*: Anushilan Acharya (anushilanacharya@gmail.com), Jakob Steiner (jff.steiner@gmail.com)

## Abstract

The cryosphere in high mountain Asia (HMA) not only sustains livelihoods of people residing downstream through its capacity to store water but also holds potential for hazards. One of these hazards, avalanches, so far remains inadequately~~poorly~~ studied as the complex relationship between climate and potential triggers is poorly understood due to lack of long-term observations, inaccessibility, severe weather conditions, and financial and logistical constraints. In this study, available literature was reviewed covering the period from the late 20th century to June 2022 to identify research and societal gaps and propose future directions of research and mitigation strategies. Beyond scientific literature, technical reports, newspapers, social media and other local sources were consulted to compile a comprehensive, open access and version controlled database of avalanche events and their associated impacts. Over 681 avalanches with more than 3131 human fatalities were identified in eight countries of the region. Afghanistan has the highest recorded avalanche fatalities (1057) followed by India (952) and Nepal (508). Additionally, 564 people lost their lives while climbing peaks above 4500 m a.s.l., one third of which were staff employed as guides or porters. This makes it a less deadly hazard than in the less populated European Alps for example, but with a considerably larger number of people affected who did not voluntarily expose themselves to avalanche risk. Although fatalities are significant, and local long-term impacts of avalanches may be considerable, so far, limited holistic adaptation or mitigation measures exist in the region. These measures generally rely on local and Iindigenous knowledge adapted with modern technologies. Considering the high impact avalanches have in the region, we suggest to further develop adaptation measures including hazard zonation maps based on datasets of historic events and modelling efforts. This should however happen acknowledging the already existing knowledge in the region and in close coordination with communities and local government and civil society stakeholders. More research studies should also be attempted to understand trends and drivers of avalanches in the region.

Keywords: Avalanches, HMA, hazards, indigenous knowledge

## 1 Introduction

High mountain Asia (HMA) comprises the largest expanse of snow, glaciers, and permafrost outside of the polar regions, and the meltwater from these cryospheric sources feeds 10 major river systems of Asia sustaining the lives and livelihoods of approximately 1.9 billion people residing downstream. As such, it is very critical in terms of ensuring water availability for
agriculture, domestic needs, industry, hydropower as well as ecosystem services (Wester et al., 2019). However, the cryosphere in HMA also holds potential for multiple hazards. Direct glacial hazards such as snow/ice avalanches, glacial lake outburst floods after calving events or high melt, glacier detachments and surges have been widely documented (Richardson and Reynolds, 2000). This is also true for indirect hazards like breaching of moraine-dammed lakes after intensive rainfall or disrupted irrigation channels associated with receding glaciers. Among all hazards, snow and ice avalanches (hereafter just
referred to as avalanches) have received little attention in scientific literature in the region.

Avalanches claim considerable worth of property and infrastructures annually and can severely affect socio-economic activities and livelihoods (McClung and Schaerer, 2006). The number of avalanche fatalities per year due to snow avalanches is estimated to be about 250 worldwide. In Europe and North America alone, avalanches claimed the lives of about 1900 people between 2000/2001 and 2009/2010 (Schweizer et al., 2021). In Switzerland avalanches caused 37% of 1023 fatalities
associated to natural hazards between 1946 and 2015 (~5.5 fatalities year$^{-1}$), making it by far the deadliest category (Badoux et al., 2016). A later study finds a considerably higher number of >20 fatalities year$^{-1}$ between 1995 and 2014 (Berlin et al., 2019) indicating the challenge of such records, considering that (Badoux et al., (2016) find a general decrease in avalanche fatalities. In the European Alps ~100 fatalities year$^{-1}$ occurred between 1937 and 2015 (Techel et al., 2016). In the USA ~16 fatalities year$^{-1}$ occurred between 1950 and 2018 (Peitzsch et al., 2020). Especially deadly events have historically been
associated with mountain warfare, ranging from an event during an Alps traverse in 218 BC to the Alp traverse by Napoleon in the 19$^{th}$ century, fighting in the Dolomites during World War I as well as impacts along the Indo-Pakistan border (Schweizer et al., 2021). Individual death tolls of many thousands have also been recorded in more recent past in the Andes (Plafker and Ericksen, 1978; Carey, 2005). Beyond human life, avalanches can have significant impacts on ecosystems (Bebi et al., 2009), livelihoods (Carey, 2005) and infrastructure. Compared to cryosphere hazards like GLOFs (Emmer et al., 2022), avalanches
have however received considerably less attention in literature.

The earliest investigations into avalanche hazards date back to the late 19$^{th}$ and early 20$^{th}$ centuries in Europe, North America, and Japan (Ancey et al., 2005) but started much later in HMA regions (Deng, 1980; Yanlong and Maohuan, 1986, 1992). Prediction of snow cover and snow avalanche behaviour is very crucial in addressing adaptation, risk management, and resilience strategies (Hovelsrud et al., 2018). In Switzerland, Bois et al., (1975), first attempted to develop models for avalanche
forecasting, and Buser (1983, 1989) further worked on developing statistical models based on Nearest Neighbors methods (NN) for local forecasting. Numerical one-dimensional models like AVAL-1D, Voellmy-Salm or, VAlanghe RAdenti (VARA)

(Barbolini et al., 2000; Bartelt et al., 1999; Christen et al., 2002) were used extensively in Switzerland and elsewhere in Europe, America and Asia. Their limitations in open terrain and terrain with several possible flow channels led to two-dimensional and three-dimensional models like SAMOS, a model developed to simulate dry snow avalanches (Sampl and Zwinger, 2004) or

RAMMS, a model to simulate flowing snow avalanches in complex terrain (Christen et al., 2010). Many of these modelling studies were motivated by risk assessments, especially after big avalanche events that caused considerable damage (e.g. the case of Galtür in 1999; Bründl et al., 2004; Fuchs et al., 2004; Keiler, 2004). As another direct result, mitigation and adaptation measures in Europe and North America have received considerable attention and in countries like Canada, Switzerland or Norway the link between scientific evidence and agencies responsible for avalanche forecasting and hazard evaluation has

been close. Parallel to model development, numerous studies have investigated the potential of remote sensing to detect and investigate snow avalanches (Eckerstorfer et al., 2016; Lato et al., 2012; Eckerstorfer et al., 2019; Bühler et al., 2009; Korzeniowska et al., 2017; Bühler et al., 2019). While some model studies have been attempted in the HMA in the 1990s and 2000s (Bhatnagar et al., 1994; Singh et al., 2005), advanced remote sensing of avalanches in the region remains largely unexplored.

Ice avalanches have received less attention in literature due to a less frequent and often much more remote occurrence but have nevertheless been of interest since many years (Alean, 1985; Röthlisberger, 1977). They have been specifically investigated in context with unstable ice in steep terrain in the European Alps, with an interest in characterizing their magnitude (Vincent et al., 2015; Margreth et al., 2017) as well as predicting potentially hazardous break offs. (Faillettaz et al., 2012, 2016; Pralong and Funk, 2006; Pralong et al., 2005; Caplan-Auerbach and Huggel, 2007). More recently the compound occurrence of ice

and snow (Margreth and Funk, 1999; Bartelt et al., 2016) as well as ice and rock (Lipovsky et al., 2008; Huggel et al., 2005; Shugar et al., 2021) as part of a mass movement have received more attention, often resulting in very high impact and long runout scenarios.

Beyond the focus on avalanches as hazards their response to climate change has been of interest, but so far no clear links are apparent and studies remain limited largely to the French Alps (Martin et al., 2001; Eckert et al., 2010; Giacona et al., 2021;

Eckert et al., 2013). To investigate avalanche occurrence and return periods in the past, dendrochronology has been extensively used (Casteller et al., 2008, 2011; Malik and Owczarek, 2009; Muntán et al., 2004; Tumajer and Treml, 2015).

In HMA, a comprehensive overview of the state of knowledge on snow and ice avalanches remains lacking. In this paper, we therefore review knowledge and data on avalanches in the region. We rely on all available scientific literature, media coverage as well as records of local and indigenous knowledge. We do so to (a) quantify avalanches as a cryosphere hazard in the region

for human life, livelihoods and infrastructure; (b) identify the sources of knowledge and its application for adaptation solutions, including a review of previously established programs in response to avalanches; (c) identify gaps and propose future directions of research and mitigation that should be undertaken to further strengthen the resilience of mountain communities against this hazard.

## 2 Data and Methodology

Scientific literature on avalanches in HMA is limited. This is in contrast to the frequent occurrence of the hazard and repeated concerns by local mountain populations who identify avalanches a prime hazard of concern. Furthermore, the inclusion of local or indigenous knowledge in our understanding of avalanches, their impacts and potential adaptation strategies remains rare in literature in general (Butler, 1987; Carey, 2005; Alcaraz Tarragüel et al., 2012). The recent AR6 report has highlighted the importance of local and I̶indigenous knowledge in mountain areas in our understanding of impacts and adaptation (Adler

and Wester, 2022). In this study we therefore collected data from ~~a variety of different~~ sources including peer-reviewed articles, newspaper articles, technical reports, social media as well as interaction with people in high mountain catchments in the region. The time period covered ranges from earliest records of deaths in the late 20th century to Ju~~ly~~ne 2022. Information from the internet was verified by cross-checking with multiple other websites and reports. The information gathered likely does not include all events due to generally poor documentation of hazard events in the region (Emmer et al., 2022). A database of

events (HiAVAL) with all available information regarding the individual events and their impacts is available open access and version controlled, where future updates can be traced (see the Data Availability section; Steiner and Acharya, 2022). We record events that affected high altitude climbers (at peaks above 4500 m a.s.l.) separately, with the argument that these are people who willingly expose themselves to avalanche risk. Due to the relatively high number of fatalities from high altitude mountaineering in HMA, these data would otherwise skew any statistics and conclusions drawn are different than when

speaking about impacts on civilians, exposed military personnel or tourists at lower elevations with no intentions of high-altitude mountaineering. In the data on high altitude mountaineering our focus is more on the fraction of avalanche fatalities among guests (often foreign) and employees (guides, porters; often local).

While a separation is not always strictly possible, this study will focus on snow and ice avalanches. Rock avalanches (Deline et al., 2015) and events that include a mixture of ice, snow, rock and debris (Shugar et al., 2021) have received increased

attention globally but especially in HMA in recent years. In this study we will however only consider them when the initial dominant source was~~,~~ and the initial release mechanism is related to either ice or snow (as in the case in Nepal in 2015; Kargel et al., 2016) rather than e.g. rock fracturing (as in India in 2021; Shugar et al., 2021). While not strictly avalanches, we do consider glacier detachments as a sub-class of ice avalanches in this review (Kääb et al., 2021).

An overview over the number and type of sources investigated in this study, and where the numbers on avalanche events and

fatalities come from is provided in Table 1. A full source document is provided in the online database (Steiner and Acharya, 2022).

Beyond national borders we use the HiMAP delineation for regions in HMA, but combine all subregions of the Tian Shan (inventory OBJECTIDs 1-3) to 'Tian Shan', Pamir and Alay (5 -7) to 'Pamir', Eastern (12) and Central (11) Himalaya to 'Himalaya Central', and all events from the Chinese Tibetan Plateau and its fringes (16, 21, 22) to 'Tibet'.


**Table 1: Type and number of most important sources used in this study, including the number of events and fatalities they cover. A table with all used sources is provided in the database (see Data Availability section). Peer reviewed literature includes thesis**

documents, 'Disaster reports' are online reports by agencies like UNOCHA, published after disaster events. Databases include public databases, like the Himalayan Database for high altitude mountaineering accidents as well as non-public repositories, like those kept by AKAH.

| Type of source | Number | Events / fatalities |
|---|---:|---:|
| Peer-reviewed literature | 14 | 42 / 733 |
| Technical reports | 4 | 7 / 883 |
| News reports | 33 | 35 / 394 |
| Websites | 5 | 29 / 151 |
| Disaster reports | 12 | 14 / 950 |
| Oral sources | 5 | 6 / 45 |
| Databases | 3 | 743 / 539 |

## 3 Avalanche types and triggering mechanisms

Currently, no documentation of the specific frequency distribution of avalanches exists for HMA nor does a systematic categorization of their type. Snow avalanches are characterized based on their release type, sliding surface, motion, and free water content. They are broadly classified into loose and slab avalanches, and both of these can be wet, dry, and damp, and can be either grounded, aerial or mixed. In glaciated, steep high mountain environments the occurrence of hanging glaciers and seracs adds other potential processes. As parts of ice collapse and drop over headwalls often more than 1 km in height the ice and snow mix turns into a powder cloud that travels at high speed and has large impact potentials (Margreth et al., 2017; Pralong and Funk, 2006). As a consequence, such glaciers are also referred to as avalanching glaciers.

### 3.1 Snow avalanches on the Tibetan Plateau and the Chinese Tian Shan

General climate conditions of the avalanche areas are considered to describe snowpack conditions of the mountain areas. Three types of snow climate in mountain ranges have been categorized; maritime, transitional, and continental (Mock and Birkeland, 2000; Yanlong and Maohuan, 1992). A number of studies on the Tibetan Plateau have investigated such classifications. In China avalanches may occur throughout the year in an alpine zone with perennial snow cover, mostly wet avalanches in summer and dry avalanches in winter. Intermediate zones of mountains in the Tibetan Plateau with ample solid precipitation and low-temperatures favor avalanches during the winter to spring season, and in some places from late autumn to early summer (Yanlong and Maohuan, 1986). Continental-type avalanches are common in Tian Shan and Altay Shan (Yanlong and Maohuan, 1986; Hao et al., 2021a). In south-eastern Xizang (Tibet) maritime avalanches are common and transitional-types over the northern slope of Daxue Shan in Sichuan, the western Nyainqentanglha Shan and Gandise Shan in southern Xizang (Yanlong and Maohuan, 1986). Similarly, Hao et al. (2021a) classified mid-latitude regions of the Central Tian Shan to be of the continental snow climate type.

Early studies along highways built on the Tibetan Plateau in China found hazardous large avalanches in the South East to be mostly wet snow avalanches with occasional full depth avalanches and traces of fresh dry snow avalanches (Yanlong and Maohuan, 1986, 1992), but release mechanisms were not discussed. Avalanches occur mainly in February and March when both temperature and snowfall increase sharply. Avalanches were generally recorded when mean daily temperatures were between 2 to 0 °C, with successive heavy snowfalls or when mean monthly air temperatures are above -3°C. Similarly, a deep snowpack (>0.7 m) on steeper slopes (>35°) favors full depth avalanches (or what the authors call depth-hoar avalanches) in the same region (Wang, 1988).

Hao et al. (2021a) provided a comprehensive study from a research station in the Tian Shan, relying on field data between 2015 and 2019 along the National Road (G218) that often gets affected by avalanches in the spring and winter. They categorized avalanches into four categories: full-depth dry snow avalanche (FDA), full-depth wet snow avalanche (FWA), surface-layer dry snow avalanche (SDA), and surface-layer wet snow avalanche (SWA) based on failure layers and the properties of snow and avalanche deposits. They identified SDAs to be the most hazardous types, occurring in the middle of the snow season, triggered by heavy snowfall and wind. They caused the highest avalanche volumes as a result. FWAs occurred late in the season as temperatures were rising quickly. In the same region Hao et al. (2018) found avalanche release prevalent when precipitation was >20.4 mm during a single storm period during the study period from November 2011 to April 2017. Of 37 avalanches, 18 were directly associated to precipitation as a trigger, followed by 10 events that were associated with rising temperatures as the main cause, whereas 6 events were associated with earthquakes with magnitudes $3.3 \leq ML \geq 5.1$ and 3 events were triggered by other factors. Heavy snow precipitation followed by rising temperatures within 3 days and daily mean temperature approaching 0.5-°C the next day tends to increase the risk of avalanches in late February to mid-March. Generally, avalanches release at slopes between 30 and 50° (Schweizer and Jamieson, 2003; McClung and Schaerer, 2006). Release zones of two thirds of mapped avalanches in Hao et al. (2018) (n=64) were between 36° and 42° steep, ($\mu$=37.2°), and even shallower in a regional study using remote sensing data, with slopes predominately between 25° and 35° (Yang et al., 2020). This is considerably shallower than release areas in the Himalaya or than in previous studies in the Alps (38.8°) or Japan (41.2°). The density of the deposits of dry avalanches ranged from 150 kg m$^{-3}$ to 300 kg m$^{-3}$, whereas wet avalanches which mostly occurred when the temperature rose sharply had a deposit density of 500 kg m$^{-3}$ ~ 700 kg m$^{-3}$ (Hao et al., 2018). More recently a study at one of these research sites showed that size and number of avalanches between 1968 and 2021 increased albeit insignificantly, following an increase in high intensity snowfall events (Hao et al., 2023). They also found an indication that during this time period snow avalanches now occur approximately one week earlier than before.

### 3.2 Snow avalanches in the Himalaya

In the Western Himalaya Sharma and Ganju (2000) classified snow climate as the Lower Himalayan Zone or Subtropical Zone (3200 – 4100 m a.s.l.), the Middle Himalayan Zone (3500 – 5300 m a.s.l.), and the Upper Himalayan Zone or High Latitude Zone (5000 – 5600 m a.s.l.). They used this to categorize avalanching based on terrain parameters together with 11-year meteorological datasets. The Lower Himalayan Zone is subject to high avalanche activity due to warm temperature, high

precipitation and relatively short cold seasons resulting in slab avalanches in peak winter, and ~~with~~ wet avalanches in late winter often mixed with debris and soil. The Middle Himalayan Zone is mostly affected by shallow slab avalanches, massive slab avalanches from wind-loaded slopes, and wet avalanches in April and May with incidences of loose snow avalanches. In a study in the Western Himalaya, 42% of avalanches were found to be loose snow avalanches and failure of soft slabs accounted for 17% (Ganju et al., 2002). In the Himalayas, avalanche formation zones are usually above the tree line at very high altitudes (above 4000 m a.s.l.) and on critical slopes (30° to 45°) where heavy snow deposition due to the wind is prevalent. Similarly, airborne powder avalanches are also common (Rao, 1985). Several studies in the Indian Himalayas conclude that release areas lie between 25° to 40° (Kumar et al., 2017; Sharma and Ganju, 2000; Singh and Ganju, 2008; Singh et al., 2018; Snehmani et al., 2014).

Most of the avalanche events in the Central Himalaya~~n region~~ occur during the drier periods due to higher temperature variability in pre-monsoon, ~~-~~making it favorable for wet avalanches ~~in this season~~. ~~W~~and ~~w~~eakened snowpack from the warm monsoon season, formed as the increased water content breaks down bonds between snow grains, results in avalanches during post-monsoon. In the western part of the HKH, most of the avalanche events occurred during the winter season as a result of excessive snow loading on snow covered terrain because of heavy winter precipitation due to Westerly disturbances (Prakash et al., 2014). For the European Alps, Ancey (2001) suggests an accumulation threshold of 30 cm day$^{-1}$ results in widespread avalanching. Beyond an accumulation of 1.5 m during snowstorms, numerous avalanches were observed in the Western Himalayas (Sharma and Ganju, 2000). Two notable avalanche events in HMA were triggered by seismic movement, namely the avalanches during the 2015 Gorkha earthquake (Fujita et al., 2017; Kargel et al., 2016) and the Lenin ~~P~~peak avalanche in 1990, although the initial source of these events was failure of glacier ice, possibly exacerbated by excessive snow loading.

A number of events that caused severe impacts happened in the Central Himalaya when monsoon precipitation came especially late (e.g. post Monsoon 2022) or typhoons in the Bay of Bengal (e.g. post monsoon 2014; see Norris et al., 2015; Cannon et al., 2017). These weather systems caused excessive snowfall that led to avalanches in the short (Figure 1; see also news reports from the post monsoon 2022, which left at least 30 and post monsoon 2014, which left at least 40 people dead in Nepal and India) as well as the long run (Fujita et al., 2017).

On a much larger temporal scale, Ballesteros-Cánovas et al. (2018) find that in the ~~W~~western Himalaya~~'s~~ warming is responsible for the increasing number of avalanches. Mostly rising temperatures during winter and early spring favored the wetting of snow and release of wet snow avalanches whose runout can reach up to subalpine slopes, and pose threats to livelihoods, infrastructures and ecosystems.

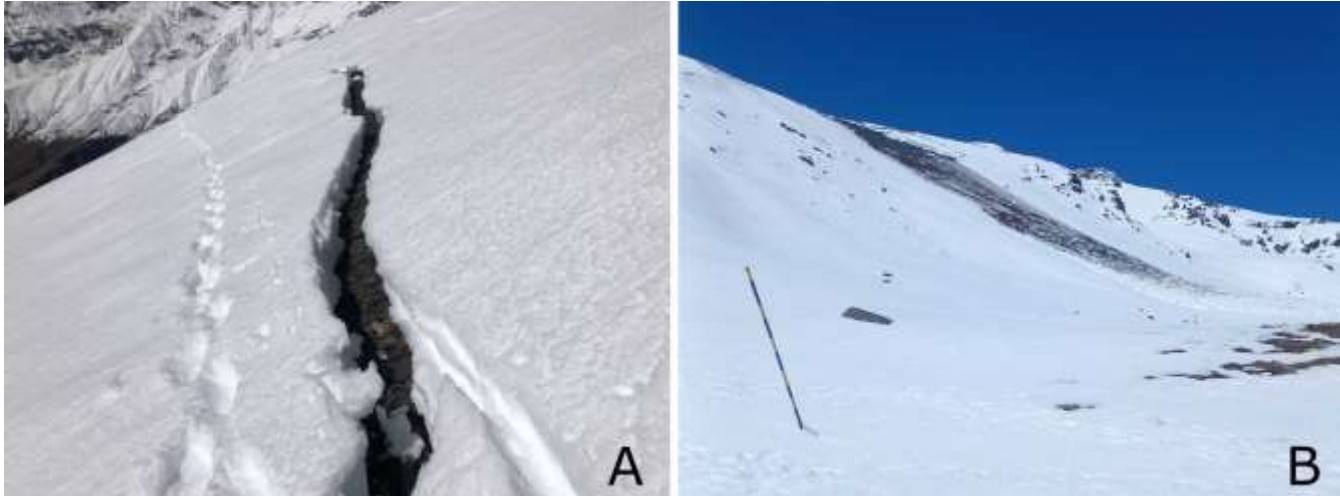

**Figure 1: (A) Glide crack in the snowpack (d~40 cm) propagating to the ground in a 20-25° slope, as a potential precursor to a glide avalanche (October 2022, Thapa Khola, Mustang, Nepal, 4900 m a.s.l. SE exposition; Pasang Tshering Sherpa). The snowpack at this location is only 1 week old after an exceptional late monsoon precipitation event, combined with still relatively high temperatures. (B) Ground avalanche at same elevation but W exposition (October 2022, Jakob Steiner).**

220

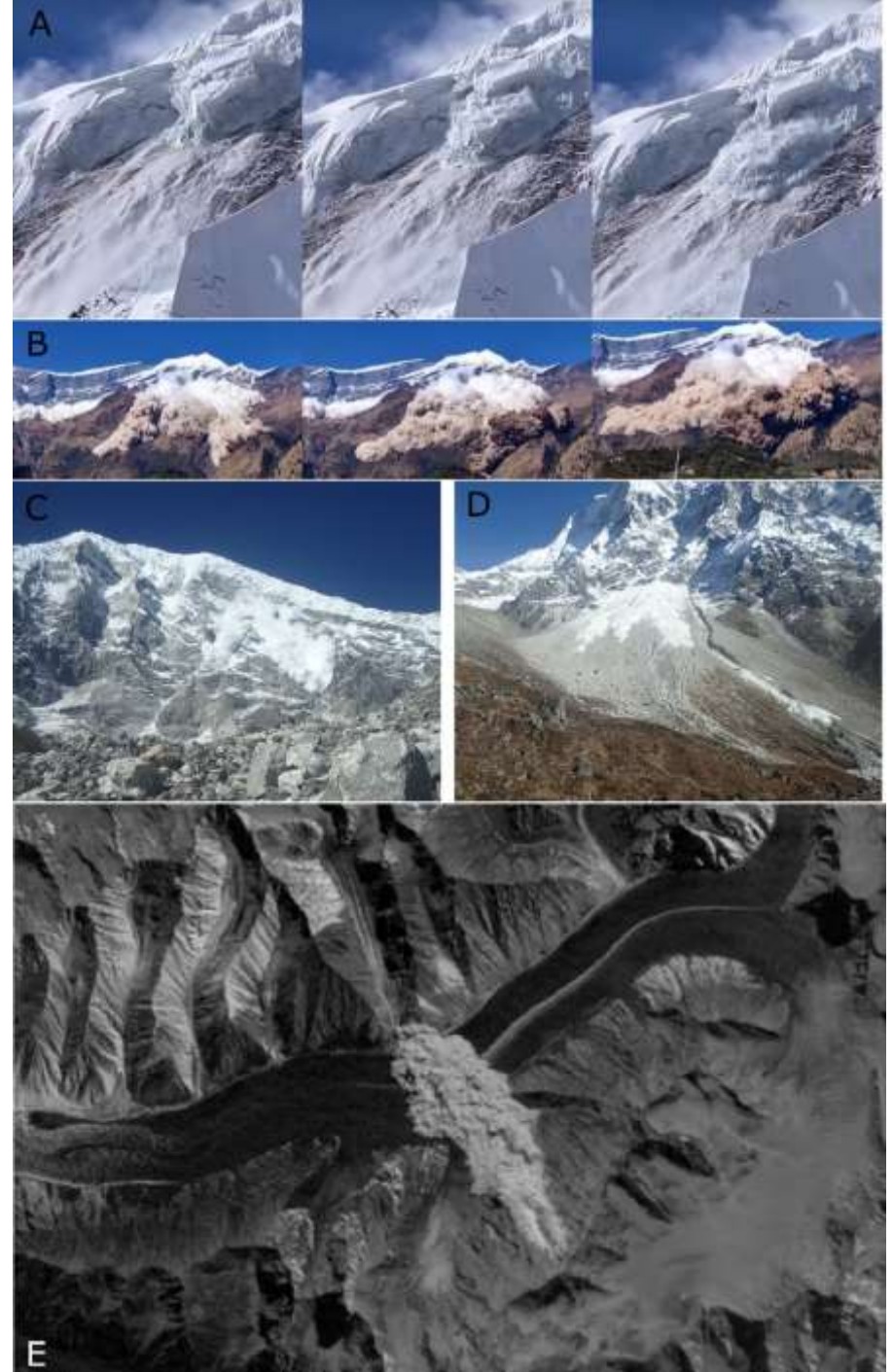

**Figure 2: (A) Serac fall at Manaslu (Nepal, stills from a movie; October 2022, Phurba Sonam Sherpa, https://www.instagram.com/p/Ci33N2lpfmW/?hl=en). (B) Kobang avalanche - stills from a mobile movie taken from Kobang Village towards Tukuche Peak (Mustang, Nepal, November 2021; https://www.youtube.com/watch?v=dxWp1D44_v4~~https://www.youtube.com/watch?v=KEeIVaLbqpg~~). Note the dimension of the**

avalanche towards the second arm on the left. The powder cloud travelled nearly 10 km along the slope, spanning an elevation of nearly 4000 m a.s.l. (C) Serac fall and subsequent power cloud at Langtang Lirung (Langtang Valley, Nepal; November 2021, Jakob Steiner). (D) Ice and snow avalanche deposits below Langshisha Ri (November 2021, Jakob Steiner). Note the gully erosion caused by repeated avalanche, and subsequent small debris flow, runouts over many seasons. (E) Powder avalanche captured on April 17, 1995 on Landsat 5 over Khurdopin Glacier. The total length of the power cloud is just above 7 km along the slope at this moment.

### 3.3 Ice avalanches

Ice avalanches are common in glaciated mountain ranges and have resulted in avalanche disasters including the Huascaran avalanches in Peru in 1962 and 1970 and the Mattmark/Allalin disaster in Switzerland in 1962 (McClung and Schaerer, 2006), events that resulted in 22088 deaths and heightened awareness of this hazard at the time. Ice avalanches can be divided into two classes, namely collapsing hanging glaciers or avalanching glaciers (Faillettaz et al., 2012) and glacier detachments (Kääb et al., 2021). The first refers to the calving/falling of ice masses or seracs at positions where glacier ice flows over a steep slope or along a ridge. This occurs as a result of a tensile failure of the ice induced by the combination of internal flow (creep) and sliding of the glacier over its bed. Multiple external factors can contribute to such an event, including melt water as well as extra load on top of the ice (Faillettaz et al., 2012). The second case results when a large piece of the glacier tongue detaches to form an ice avalanche.

In HMA both types have been frequently reported. Collapse of glacier seracs in icefalls are responsible for many fatalities in the HMA. For instance the 2014 Khumbu ice fall avalanche claimed 16 lives, and again 23 in 2015 (Thakuri et al., 2020). In 2008, serac falls at the bottle neck below the peak claimed 11 lives on Chhogori (Kutsch et al., 2017). Seismic activity can also induce failure. The avalanche in Langtang Valley in 2015 was triggered by the Gorkha earthquake (Kargel et al., 2016) and the same earthquake was responsible for the avalanche in the Khumbu ice fall in 2015 (Bartelt et al., 2016). The ~~so called~~so-called Lenin ~~P~~peak disaster that killed 43 people ~~on Mount Lenin~~ on 13 July 1990 was also caused by a serac fall after seismic shocks. Ice avalanches from seracs or hanging glaciers can occur multiple times a day at the same location, resulting in runouts and powder clouds of just a few hundred meters to multiple kilometers (Figure 2). Ice avalanches also provide a link between the accumulation and ablation zones in the steep topography of the region (Laha et al., 2017). Ice avalanches are more frequent in spring or summer as more basal melt water becomes available (McClung and Schaerer, 2006). Some ice avalanches, like the case in Kobang in 2021 (Figure 2B) or the Langtang avalanche in 2015 also increase their downstream impact as they pick up more snow and debris along their path and hence become classical examples of cascading hazards.

Glacier detachments are considerably rarer than collapses of hanging glaciers, but recent studies attribute previously unknown cases to this process (Kääb et al., 2021; Leinss et al., 2021). 140 people lost their lives in 2002 Kolka–Karmadon rock-ice avalanche (Huggel et al., 2005) and 9 people got killed in the Aru Co detachment (Kääb et al., 2018). Two cases that occurred in 2022 with a lot of media attention in Italy and Kyrgyzstan, possibly fall in one of these two types or somewhere in between. Similarly, previous events in the Upper Indus Basin, that caused significant repeat destruction has now been identified as a process where melt water accumulates in crevasses and leads to a failure of the tongue, rather than a GLOF as previously suspected.

## 4 Recorded snow and ice avalanches in HMA

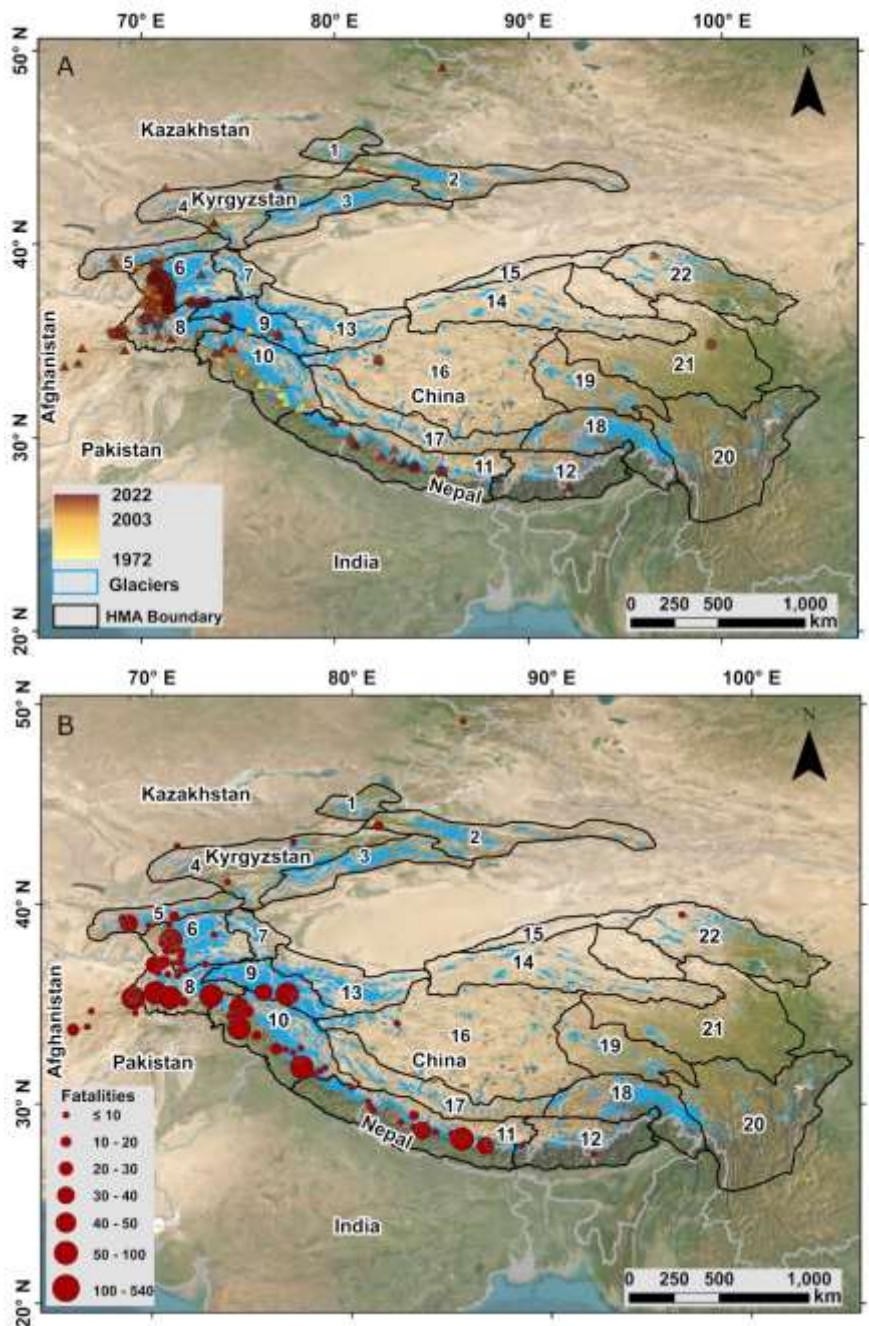

Figure 3. (A) Distribution map of avalanche events throughout HMA since 1972. Locations marked with a triangle are events with some recorded impact (fatalities or injured people, livestock killed or infrastructure damaged or destroyed). The colour refers to the year of the record. (B) All events with recorded fatalities from individual events. The black outline shows regional outlines of High Mountain Asia, with numbers referring to region-IDs (Bolch et al., 2019) used in the avalanche database.The black outline shows the regional outline of high mountain Asia, with IDs reflected in the avalanche database (Bolch et al., 2019).

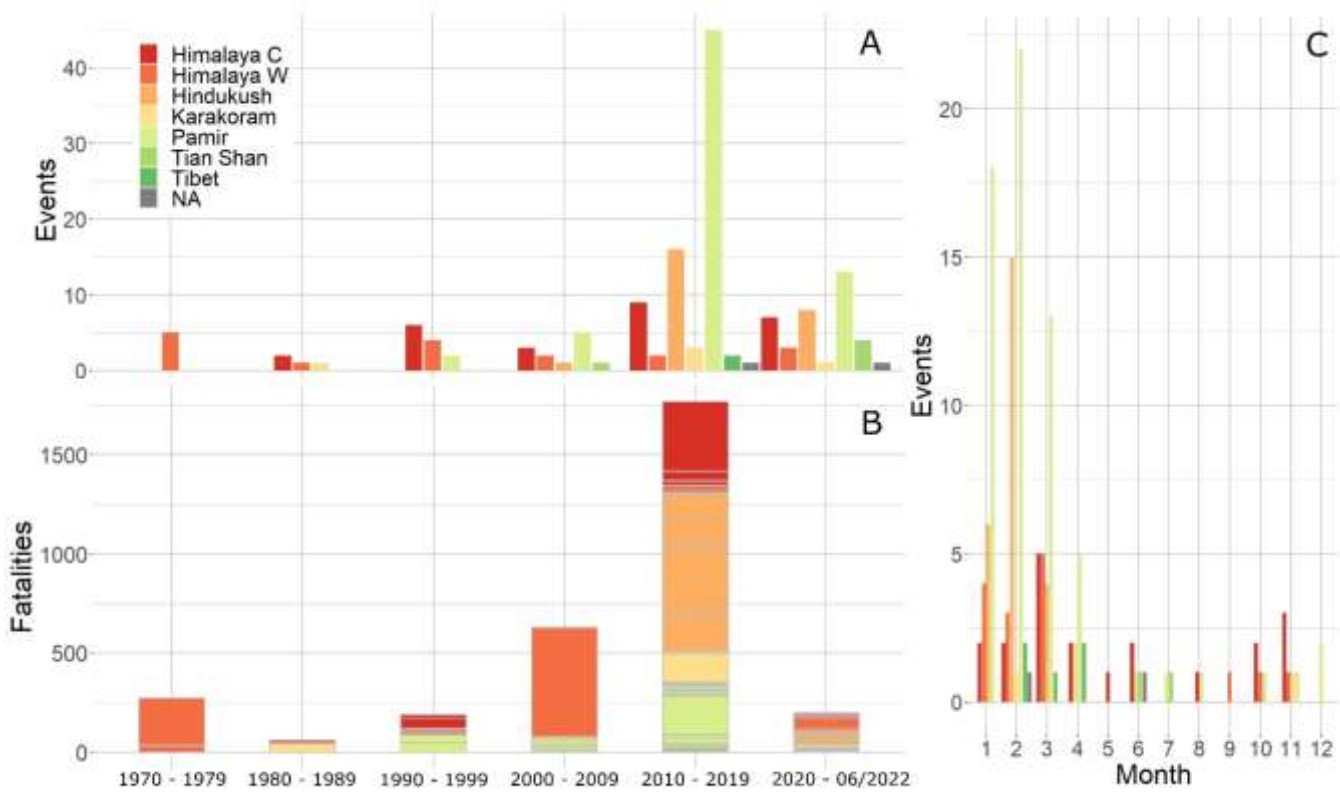

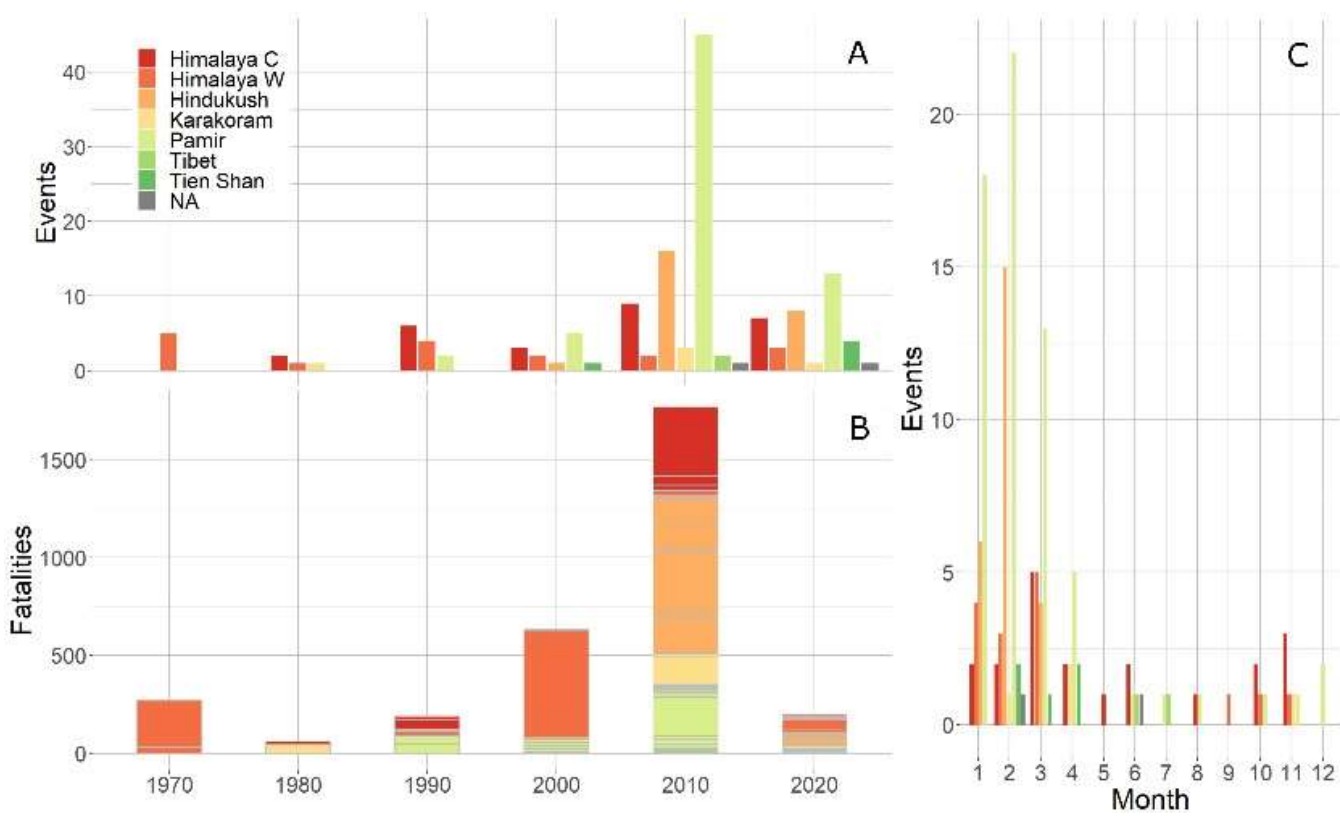

**Figure 4. (A) Decadal avalanche events with some recorded impact (fatalities or injured people, livestock killed or infrastructure damaged or destroyed, ~~wide bars~~) and (B) fatalities (stacked ~~thin~~ bars, where grey borders ~~separates~~separate individual events). Note that the decade from 2020 only includes two and a half years of data. (C~~B~~) Avalanche events with recorded impacts by month. Note that for the months from May to December a maximum of 3 events per region and month have been recorded, all happening in different years respectively.**

**Table 2. Overview over avalanches with more or equal to 10 fatalities.**

| Location | Year | Month | Day | Latitude | Longitude | Country | Type | Fatalities |
|---|---|---|---|---|---|---|---|---|
| Lah~~auual~~aul and Spiti | 1978 | 3 | NA | 32.759515 | 76.19334 | India | NA | 30 |
| Lahaul Valley | 1979 | 3 | NA | 31.841332 | 77.485476 | India | NA | 237 |
| Shigar | 1988 | NA | NA | 35.592339 | 75.568032 | Pakistan | NA | 42 |
| Gokyo | 1995 | 11 | 12 | 27.912404 | 86.683499 | Nepal | NA | 48 |
| Anzob | 1997 | 10 | 23 | 39.041862 | 68.836962 | Tajikistan | NA | 46 |
| Banihal, Jammu and Kashmir | 1998 | 2 | 25 | 33.430363 | 75.214877 | India | NA | 11 |
| Jammu and Kashmir | 2005 | 2 | 18 | 33.793152 | 74.346548 | India | snow avalanche | 540 |
| Jirgital | 2006 | 1 | 30 | 39.405905 | 71.091765 | Tajikistan | snow avalanche | 18 |
| Anzob | 2007 | 12 | 22 | 39.272888 | 68.541344 | Tajikistan | snow avalanche | 16 |

| Ili | 2008 | 3 | 17 | 43.92 | 81.32 | China | NA | 16 |
|---|---|---|---|---|---|---|---|---|
| Salang | 2009 | 1 | 16 | 35.305848 | 69.049623 | Afghanistan | NA | 10 |
| Salang | 2010 | 2 | 8 | 35.310158 | 69.045199 | Afghanistan | NA | 175 |
| Kohistan | 2010 | 2 | 16 | 35.411909 | 72.94035 | Pakistan | snow avalanche | 120 |
| Afghanistan Fayzabad | 2012 | 1 | 18 | 37.1175 | 70.579722 | Afghanistan | snow avalanche | 29 |
| Afghanistan | 2012 | 1 | 23 | 36.949934 | 70.116616 | Afghanistan | snow avalanche | 43 |
| Badakhshan | 2012 | 3 | 4 | 38.2 | 70.933333 | Afghanistan | snow avalanche | 196 |
| Gayari Military Base, Ghanche | 2012 | 4 | 7 | 35.498008 | 76.753358 | Pakistan | ice and rock avalanche | 140 |
| Kaicheng Town, Yuanzhou District | 2012 | 6 | 5 | 36.210025 | 104.213076 | China | NA | 10 |
| Kaj-Lar, Raghistan, Badakhshan | 2015 | 2 | 21 | 37.702133 | 70.735718 | Afghanistan | snow avalanche | 12 |
| Langtang | 2015 | 4 | 25 | 28.237596 | 85.498829 | Nepal | ice and rock avalanche | 350 |
| Siachen Glacier | 2016 | 2 | 3 | 35.452442 | 77.085467 | India | NA | 10 |
| 2016 Karnali avalanche | 2016 | NA | NA | 29.43017 | 83.095098 | Nepal | NA | 16 |
| Gurez Sector | 2017 | 1 | 25 | 34.634041 | 74.773255 | India | NA | 24 |
| Nuristan | 2017 | 2 | 4 | 35.317202 | 70.898674 | Afghanistan | NA | 137 |
| Barghud, Maimai, Badakhshan | 2017 | 2 | 4 | 38.155406 | 71.2398802 | Afghanistan | snow avalanche | 10 |
| Line of Control | 2020 | 1 | 14 | 34.397451 | 73.860665 | India | snow avalanche | 10 |
| Line of Control | 2020 | 1 | 14 | 34.827012 | 74.358551 | Pakistan | snow avalanche | 55 |
| Daykundi | 2020 | 2 | 13 | 33.703958 | 66.036936 | Afghanistan | NA | 22 |
| Zarandabad, Khahan, Badakhshan | 2021 | 3 | 5 | NA | NA | Afghanistan | snow avalanche | 14 |
| Salang | 2021 | 3 | NA | 35.310158 | 69.045199 | Afghanistan | NA | 14 |
| Chamoli | 2021 | 4 | 23 | 30.822323 | 79.966271 | India | ice avalanche | 15 |
| Kunar Province, Afghanistan | 2022 | 2 | 7 | 35.117071 | 71.559069 | Afghanistan | snow avalanche | 19 |

The HiAVAL database has a total of 681 entries between 1972 and 2022. Only one event has an unclear year of occurrence, 97% have a known month and 95% a known day. For 91% of the events the avalanche type is recorded. For 81% the record just notes a slab avalanche and another 3% each note a snow or wet snow avalanche. Less than 0.5% are ice avalanches and

285 2.5% glacier detachments. For 97% of all events coordinates are known, however in nearly all cases that only refers to the

general mountain slope and does not allow to identify source or runout area exactly. 91% of all avalanche events occur between January and March (76% if only considering events with recorded impacts; Figure 4B).

Only 145 (21%) of all events have some reported impact and in the discussion of the results we will only refer to this fraction (Figure 4). A large number of avalanches without any recorded impact were mapped in Afghanistan, where a dedicated monitoring project has been put in place after 2015. 44% of these events occurred in the Pamir, 18% in the Central Himalaya, 17% in the Hindukush, 11% in the Western Himalaya and 3% each in the Karakoram and the Tian Shan and less than 1% on the Tibetan Plateau. 48% of these events with impacts were recorded in Afghanistan, 16% in India, 15% in Nepal, 9% in Tajikistan, 4% in Pakistan, 3% each in Kazakhstan and Kyrgyzstan and 2% in China. No recorded events exist for Bhutan. There is a clear increase in events after 2010, which can be explained by the monitoring project initiated in Afghanistan that comprehensively recorded events in the Hindukush and the Pamir. Fatalities are also extreme in this decade, which can be explained by extreme events in both Nepal and Afghanistan in 2015 (Table 2). In 116 events a total of 3131 people were killed, 2519 of which in Afghanistan (1057), India (952) and Nepal (508). In 45 events a total of 531 injured people were recorded. In 24 avalanche events a total of 1902 livestock were recorded killed, often with more than 100 hundred individual yaks or goats killed in a single avalanche.

Avalanches with recorded impacts have been documented in nearly all parts of HMA, naturally clustered around the topographically rougher ring of mountain ranges from the Tian Shan in the North West, to the Pamir, Hindukush, Karakoram and Himalaya in the South East and virtually no records on the Tibetan Plateau (Figure 3Figure 3A). Large scale events with more than 100 fatalities have occurred all over the Himalayan arc and the Pamir (Figure 3Figure 3B). In 32 avalanches more than 10 people were killed, with 2435 fatalities making up 78% of the total (Table 2).

Avalanche fatalities for single events are generally high ($\mu = 27$, $\sigma = 32$). 2% (11 events) of all events had more than 50 fatalities and 12% events had less than 10 fatalities. However, in many years there were no recorded events at all. Recorded events increase notably after 2010.

In Afghanistan and Tajikistan predominately civilians were affected in their respective settlements or during travels on foot or the road. Reported cases in China and Kyrgyzstan often included workers in infrastructure projects. Relatively few events impacted tourists not involved in high altitude mountaineering (which we discuss separately in section 4.3). A total of 201 military personnel (soldiers as well as civilian employees of the army) were killed along borders in the region, predominately in India and Pakistan.

**4.1 Central Asia (Pamir, Hindukush, Tian Shan, Inner Tibet; Kazakhstan, Kyrgyzstan, Tajikistan, Afghanistan, Pakistan, China)**

Avalanches in the Central Asian Tian Shan and Pamir (and here mainly limited to Kyrgyzstan and Tajikistan, with fewer cases in Kazakhstan and China) have affected skiing areas, blocked roads, and destroyed infrastructure like bridges, schools, hospitals, and buildings and have repeatedly caused loss of life. More than 270 people were killed since the 1990s (see selected events in Table 2). Hotspots were roads like the Anzob Ppass between Dushanbe and Khujand, where 77 people died. As a

result, preliminary assessments have been conducted. 298 avalanches have been recorded in Kyrgyzstan between 1990 and 2009 and an initial susceptibility mapping was conducted for one of the most important roads in the country (Chymyrov et al., 2019). The only country in HMA where avalanche deaths were recorded in a skiing area is Kazakhstan, where two accidents happened in the Shymbulak resort.

Afghanistan has seen especially strong avalanche impacts in recent decades. Nearly 1000 people were killed between 2009 and 2022 alone. With events occurring in very remote regions, relief operations are often difficult. A recent remote sensing study suggests that since 1990 approximately 800 000 large avalanches with an area coverage of 28 500 km$^2$ occurred in Afghanistan since 1990, and are responsible for damaging several villages and blocking roads and 19 streams. The avalanche deposits have impacted about 26.2% of the river network in the high mountains in the Amu Panj basin (Caiserman et al., 2022). Hazard and vulnerability risk assessments in three countries (Tajikistan, Afghanistan, and Pakistan) showed that 571 villages are at risk of avalanches, 30 000 people directly being exposed to avalanches, 75% of which in Afghanistan (Chabot and Kaba, 2016). A World Bank report (World Bank, 2017) on hazards in Afghanistan notes that 2 million people, $4 billion of assets, and 10 000 km of roads are exposed to avalanches, of which $990 million in Badakhshan alone. From 2000 to 2015, more than 153 000 people were affected. More than 300 people were killed in 2015, the majority in Panjshir Valley. No significant efforts in terms of mitigative measures for protecting settlements and infrastructures have been made so far except few attempts to test early warning systems around the Salang Pass.

**4.2 Karakoram and Himalaya (India, Nepal)**

Ganju and Dimri (2004) provide an overview of avalanches in the Indian Western Himalaya including 109 villages in Himachal Pradesh, 91 in Jammu and Kashmir, and 16 in Uttarakhand. The avalanches created sudden cut-off in essential supplies, loss of forest cover, loss of soil cover, road damage, hindrance in the hill development schemes, and deposits resulting in dams subsequently causing flooding. Moreover, these avalanches severely affected military operations and border area security. Many important road connections, such as Manali - Leh, and Hindustan - Tibet road in Himanchal Pradesh remain cut off for at least five to six months a year due to winter snow and avalanche threats. A case study in Jammu and Kashmir, India, from February 2005 reports the incidences of snow avalanches in the Anantnag, Poonch, Doda, and Udhampur districts (Uttarakhand Open University, 2017). More than 200 000 people were directly affected by a series of avalanches and 278 people lost their lives and avalanches submerged several houses.

Avalanches with recorded impacts in the Central Himalaya are few and are mainly associated to extreme weather events, especially when typhoons from the Bay of Bengal result in intense snowfall. Such events have resulted in countless avalanche deaths every decade since the 1970s. Avalanches released during the 2015 earthquake also caused numerous deaths (>350). Although the main trigger in this case was the seismic shock, heavy snowfall from the previous year's typhoon potentially exacerbated the impact (Fujita et al., 2017).

 **4.3 Avalanche fatalities in high altitude mountaineering**

HMA is home to the highest peaks on the planet, including all above 8000 m a.s.l. High altitude mountaineering is an important economy in some regions (India, Nepal, China and Pakistan; to a more limited degree in the Central Asian states), which started with the first attempts of peaks above 6000 m a.s.l. at the beginning of the 20[th] century. In areas like the Khumbu in Nepal or the Baltoro region in Pakistan individual peaks are scaled by more than 500 people a year. Table 3 depicts the avalanche deaths on all the fourteen 8000 m a.s.l. peaks (covering a period from 1895 to 2021, although most cases stem from the time after the 1930s) as well as high peaks for which accident records exist in the Himalayan Database (for the Himalaya, recording from 1930 to 2021) or news articles (outside of it). Deaths because of serac falls, icefalls and snow avalanches are incorporated in this table. Especially deadly events include an avalanche on ~~Mt.~~ Qomolongma on April 18, 2014, claiming the lives of 16 and an earthquake-triggered avalanche on April 25, 2015 killing 23, halting the expedition season for two consecutive years (2014 and 2015).

**Table 3. Avalanche accidents and associated fatalities on high altitude peaks in HMA between 1895 and 2022.**

| Peak | Elevation | Country | Accidents | Fatalities | Hired staff | Hired staff [%] |
|---|---|---|---|---|---|---|
| Others | - | - | 56 | 207 | 49 | 23.7 |
| Xixapangma | 8013 | China | 8 | 15 | 1 | 13.3 |
| Gasherbrum II | 8035 | Pakistan/China | 3 | 4 | 0 | 0.0 |
| Faichan Kangri (Broad Peak) | 8047 | Pakistan | 4 | 5 | 1 | 20.0 |
| Gasherbrum I | 8068 | Pakistan/China | 4 | 10 | 4 | 40.0 |
| Annapurna I | 8091 | Nepal | 19 | 41 | 13 | 31.7 |
| Diamir (Nanga Parbat) | 8125 | Pakistan | 11 | 36 | 11 | 30.6 |
| Manaslu | 8163 | Nepal | 13 | 41 | 14 | 34.1 |
| Dhaulagiri | 8167 | Nepal | 16 | 42 | 18 | 42.9 |
| Cho Oyu | 8201 | Nepal/China | 4 | 9 | 5 | 55.6 |
| Makalu | 8463 | Nepal/China | 3 | 3 | 1 | 33.3 |
| Lhotse | 8516 | Nepal | 9 | 13 | 3 | 23.1 |

| Kangchenjunga | 8586 | Nepal/India | 4 | 8 | 2 | 25.0 |
|---|---|---|---|---|---|---|
| Chhogori (K2) | 8611 | Pakistan/China | 10 | 24 | 4 | 16.7 |
| Qomolongma (Mt. Everest) | 8848 | Nepal/China | 33 | 106 | 70 | 66 |
| **TOTAL** | | | **197** | **564** | **191** | 33.9 |

Overall, out of a total of 564 avalanche deaths, 191 deaths (34%) were of locally hired personnel, with the highest fraction (66%) on the highest peak, observations that have already been made in a previous study exclusively focusing on avalanche deaths during climbing on the highest peaks (McClung, 2016). These rates are more concentrated in the Central Himalaya, where expeditions rely more on large groups of porters and guides. In the Khumbu, more locally hired people are affected by avalanches as they contribute to a larger extent and for a longer time period to prepare the way to the summit (so called 'icefall doctors'), exposing themselves to avalanche-prone areas. Other mountains with peaks above 4500 m a.s.l. recorded 207 fatalities in 56 different accidents throughout the region.

## 5. Avalanches and glacier dynamics

Avalanches don't only pose direct hazards but also affect glacier mass balance and as such indirectly water resource related risks in the downstream. Several studies have shown that avalanches contribute significantly to the net accumulation in HMA. Usually, debris-covered glaciers, associated with high relief and steep headwalls, are fed by avalanches (Benn and Lehmkuhl, 2000; Laha et al., 2017; Hewitt, 2011; Azam et al., 2018). Benn and Lehmkuhl (2000) state that the avalanche contribution to the mass balance of Khumbu Glacier, Nepal, is 75%. Also large valley glaciers like Batura Glacier, Karakoram, and Fedchenko Glacier, Pamir, receive significant amounts of ice and snow by avalanches from the surrounding high relief valleys. Ragettli et al. (2015) found that the melt of avalanche deposits contributed 16% of the annual water balance in the Lirung sub-catchment in the Central Himalaya, which constitutes 43% of total snowmelt in the whole catchment. Another study on the Satopanth, Dunagiri and Hamtah Glaciers in the Western Himalaya shows that avalanches from headwalls contributed more than 95% of the total accumulation. A huge disparity was found while simulating mass balance of these glaciers with (-0.2 ± 0.5, -0.3 ± 0.2, and -0.5 ± 0.3 m w.e. a$^{-1}$) and without consideration (-2.0, -1.0, and -1.5 m w.e. a$^{-1}$) of avalanches (Laha et al., 2017). Glaciers that don't show signs of rapid retreat or considerable downwasting even if the accumulation area ratio (AAR) of these glaciers is less than 0.2, are likely strongly avalanche-fed (Laha et al., 2017). In the Tian Shan, a study using high resolution imagery and modelling suggests that contribution to accumulation is considerably smaller, only up to 10% (Turchaninova et al., 2019).

## 6 Mitigation measures for avalanche hazards

Studies with~~in~~ respect to avalanche hazard mitigation have been conducted at a government level largely in the Indian Western Himalaya, with some monitoring also documented on the Tibetan Plateau in China and ~~community based~~community-based
hazard mitigation approaches in Afghanistan, Tajikistan and Pakistan.

### 6.1 Field monitoring and desktop analysis

Avalanche prediction based on ~~NN~~ Nearest Neighbors was first adopted by ~~(~~Bhatnagar et al.~~,~~ (1994) for short-term avalanche forecasts in the Indian Himalaya. A few more studies were conducted using the ~~NN~~ method in the Indian Himalaya (Singh and Ganju, 2004, 2008). Singh et al. (2005) also modified ~~the NN based avalanche~~this forecast model with a mesoscale weather
forecast, allowing for predictions up to 4 days in advance. In the Khumbu ~~region~~ (Kattelmann and Yamada, 1996) and Langtang region (Fujita et al., 2017) of the Central Himalaya avalanches have been investigated regarding their triggers but no comprehensive forecasting exists. Both optical remote sensing (Bühler et al., 2013; Kumar et al., 2017, 2018; Caiserman et al., 2022; Kumar et al., 2017; Snehmani et al., 2014; Singh et al., 2019b) and Synthetic Aperture Radar (SAR)~~SAR~~ technologies (Eckerstorfer and Grahn, 2021; Patil et al., 2020) have been used for avalanche activity detection. SAR technologies have the
advantage of being completely independent of light and weather conditions. In the Tian Shan, Yang et al. (2022) applied RS image interpretation and drone photographs together with field investigations to determine avalanche susceptibility. In the Western Himalaya, SASE (now under Defence Geo-Informatics Research Establishment/DGRE) is using RS and GIS technology and incorporates multi-spectral, hyper-spectral, ~~Synthetic Aperture Radar (~~SAR~~)~~, and various other optical and microwave satellite data to extract snow cover and terrain information from inaccessible terrains (Singh et al., 2019a). This
also included susceptibility mapping at the regional scale (Kumar et al., 2017, 2018; Singh et al., 2019a, 2018).
Gusain et al. (2019) created a comprehensive system for providing avalanche information with different levels of detail in the Indian Karakoram using the Advanced Space-borne Thermal Emission and Reflection Radiometer (ASTER) Global Digital Elevation Map (GDEM) version-2 at TIR-2 (11.50-12.51) with a spatial resolution of 30 m, as well as Cartosat-4 DEM at a resolution of 10 m, and satellite images from Landsat-8 and PLEIADES-1A. The resulting system is supported by GIS and
enables easy visualization of the terrain ~~and related information~~. Similarly, Dewali et al. (2014) designed a real-time avalanche path warning and navigation system based on a global positioning system (GPS) within a GIS environment for the visualization of terrain maps and navigation in avalanche-prone regions, specifically in the Manali-Dhundi area of the lower Himalayas. Camp locations, pedestrian routes and avalanche sites along pedestrian routes, have been digitised and past avalanche accidents along pedestrian routes, past avalanche occurrences and~~,~~ climatology of the region have also been mapped in a GIS
environment. This was paired with an online platform for geospatial data visualisation, snow status updates and avalanche forecast dissemination for Jammu and Kashmir, Himanchal Pradesh and Uttarakhand. A portable GPS-based Real-time Avalanche Path Warning and Navigation System has also been developed. SASE has a rich network of wireless sensors with 35 wireless nodes and total of 128 snow and meteorological sensors recording 14 different climatological variables in an area

covering 400 ,000 m² (2890 m to 3350 m a.s.l.) in the Pir Panjal Range in Himachal Pradesh. A database of mechanical properties and snowpack structures information of typical Himalayan snow over the last decade allows for snow cover and avalanche simulations (Singh et al., 2019a). This collective knowledge led to the establishment of an avalanche warning bulletin published regularly (https://www.drdo.gov.in/avalanche-warning-bulletin; last accessed 2023/04/22). In the Indian Karakoram, Singh et al. (2021) discussed the installation of 9 surface observatories and two automated weather stationsAWS to help in generating avalanche forecasts. Numerical models like RAMMS were applied in the Western Himalaya and Karakoram to simulate the avalanche incidents, for instance, to simulate avalanche incidents of 5th January 2018 near Chowkibal Village, Jammu, and Kashmir, India (Singh et al., 2020) and on, in Siachen Glacier (Sardar et al., 2017). In the southeastern Tibetan Plateau studies about avalanches were first carried out in 1964, with later extensive field visits between 1981-83 in the Hengduan Shan, where two avalanche observation stations, one each on Baimang-Xue mountain and Bomi mountain were established (Deng, 1980). This was more recently complemented by more detailed scientific research (Hao et al., 2018, 2021a, b).

Focus Humanitarian Assistance (FOCUS; now AKAH) is helping vulnerable communities to build resilience to disasters like avalanches in Tajikistan, Pakistan, and Afghanistan. It has established a network of 82 manual weather monitoring posts in January 2015 (17 in Afghanistan, 45 in Pakistan, and 25 in Tajikistan) to help forecast avalanches in 571 remote high-risk villages to reduce fatalities. They are continuously conducting a program for community avalanche education, and also providing a community training manual on snow avalanche risk management. Furthermore, they are involved in identifying high-risk vulnerable villages through avalanche terrain mapping, managing specialists to analyze meteorological datasets, and training the community trainers in avalanche safety and weather monitoring (Chabot and Kaba, 2016). AKAH has been conducting comprehensive training on avalanche safety to empower communities to be prepared for avalanches. For instance, in 2021, AKAH provided training to 2,002 people from 205 villages in Afghanistan, equipping them with Avalanche Preparedness Team (AVPT) toolkits. Training covers a range of skills, including data collection from weather stations, which helps in predicting the likelihood of avalanche occurrence. As a result of such trainings, 263 villages were provided with avalanche warning messages in 2021. Similarly, AKAH has conducted about 800 HVRAs (Hazard Vulnerability and Risk Assessment), together with two habitat assessments. These HVRAs then were utilized to develop nearly 700 village disaster management plans and two habitat plans (AKAH, 2023). In Badakhshan, Afghanistan, several government and non-government organizations are conducting programs for community resilience forming community emergency response teams (CERTs), avalanche preparedness teams, and school emergency response teams (SERTsS) for disaster management generally raising avalanche awareness. However, effective emergency response, rescues, and evacuation are affected by economic challenges and security issues. In at least one case, the program has also resulted in relocation of villagers, as the Verizon Village in Shughnan located in an avalanche-prone area was completely relocated to a safer location (Lashkari, 2022).

## 6.2 Structural measures

Prevention and control methods are divided into permanent (passive) and temporary (active) measures. Temporary measures are applied for short periods when avalanches are expected and are advantageous because of their flexibility and low cost but need a continuous evaluation of avalanche hazards and the application of safety measures. Permanent measures are usually expensive and indulge large engineering works, but don't need daily hazard evaluation (McClung and Schaerer, 2006). In HMA, structural measures, besides being costly, are often difficult to implement due to logistic constraints (Ganju and Dimri, 2004). A study in the eastern Karakoram found that proposed structural mitigation measures are infeasible due to steep, unstable and glaciated terrain. Slope stabilization considering preventive triggering of small avalanches has potential but necessary infrastructure and permits are difficult to obtain, install and maintain (Singh et al., 2021).

Control structure schemes are designed based on avalanche zones; formation zone control structures, middle zone control structures, and runout zone control structures. Numerous control structures in different zones were implemented in Jammu and Kashmir, Himanchal, and Uttarakhand in India (Rao, 1985; Singh et al., 2019a; DRDO, 2018). Several snow bridges and snow stakes are installed by SASE above the Jawahar Tunnel on the Jammu-Srinagar National Highway. Snow nets, snow fences, jet roofs, and baffle walls are installed along different sections of mountain roads in the Indian Himalaya as formation zone control structures and, galleries (e.g. along the Manali-Leh highway) are planned. Retarding mounds used for breaking the momentum of the avalanche mass and catch dams to retain avalanche mass in large depressions (e.g. near Shri Badrinath temple) have been installed as well (Singh et al., 2019a). Artificial triggering has been tested in Gulmarg, Jammu and Kashmir in February 2018, using rocket launchers (84mm RL) by the Indian Army (Singh et al., 2019a). AKAH has also shown some initiative in implementing tailored mitigation measures such as protective walls, tree planting, and retrofitting buildings for greater resilience in Afghanistan (AKAH, 2023).

## 7 Indigenous and local knowledge, adaptation, concerns and ~~community based~~community-based disaster management – 3 case studies

Knowledge about the cryosphere, hazards and adaptation already present in mountain areas has always been known (Fleetwood, 2022), but still remains rarely included in the development of solutions. At the same time the value and importance of Indigenous and local knowledge and adaptation solutions in mountains have been acknowledged (Adler and Wester, 2022). Below we detail three case studies on how local populations have raised their concerns about avalanches with respect to livestock (Central Himalaya/Nepal), local adaptation solutions where avalanches are turned into opportunities for other climate related risks (Karakoram/Pakistan), and community-based disaster management stemming from high vulnerability in remote areas (Hindukush/Afghanistan).

## 7.1 Avalanches and Yak herding

Yak herding is a common mountain economy, prevalent all over HMA (Wu et al., 2016). In Nepal, it is an important source of income in the Lower Mustang and as well as the Langtang Valley, both sites also research sites frequented by the authors. Avalanches have been identified as a major risk for herders in Mustang where recorded events in 1987, 1998, and 2003 killed 62, 25 and 70 yaks respectively of a single owner (Degen et al., 2007) and again 14 of multiple owners at the same location in November 2021 (Figure 2B) and 7 more in March 2022. In Langtang five killed yYak were recorded in August 1997

(McVeigh, 2004), over a thousand reportedly after co-seismic avalanches in 2015 and locals have noted avalanches as a major concern in more recent past (Tuladhar et al., 2021). While a contested Iindigenous knowledge by some villagers, locals claim that *manes* or Buddhist sacred walls (Emerman et al., 2016) have also been strategically erected at locations that are or have been avalanche runout zones in the past (their location aligning well with known avalanche locations). This is done to preserve memory on where to be careful along the path or where not to build as well as avert the potential wrath from deities expressed

through natural disasters like avalanches.

Avalanches have been identified as a threat that can wipe out livelihoods within an instant and potentially force people to move away and look for other professions (McVeigh, 2004; Degen et al., 2007). As a response, the International Centre for Integrated Mountain Development (ICIMOD) has started a community based participatory approach for avalanche hazard adaptation. This includes the preparation of participatory maps, constant exchange with communities as well as the use of satellite data to

visualize potential of monitoring (Eckerstorfer and Grahn, 2021).

In the Central Himalaya, no other mechanisms to deal with avalanche hazards have been discussed so far. Efforts generally deal with the immediate impacts caused by the avalanche hazard. For instance, the Conservation Area Management Committee (CMC) of the Annapurna Conservation Area Project (ACAP), provided NPR 1000 per yak (~7.5 USD; approximate value of a fully grown yak at the time of writing is 150 000 NPR) to every owner that lost their animals in an avalanche in Kobang in

November 2021 and March 2022 (personal communication).

## 7.2 Avalanches and irrigation

High altitude irrigation of agricultural land is challenging as irrigation channels are difficult to upkeep. Such channels, referred to as *khul*, are commonly maintained in the Hunza Basin. However their maintenance under the threat of avalanches is challenging (Ashraf and Ahmad, 2021; Ashraf and Akbar, 2020). Slightly further south in the Rupal Valley, such channels

source melt water directly from locations with repeat avalanche deposits, as they are otherwise disconnected from the Chungphare Glacier (Nüsser et al., 2019). In the region, the practice of glacier marriages (or glacier growingcrafting) is also practiced as a response to water scarcity (Tveiten, 2007; Faraz, 2020). A complex and long-standinglong-standing practice combining local myths and a precise knowledge of the local cryosphere, glacier ice from lower elevations is placed in caves in permafrost environments at higher elevations, exposed to avalanche activity. This way the ice deposits grow and are

eventually harvested for irrigation water later.

A similar approach, although less stooped in local folklore, is being pursued in Baltistan. In a locality that has been heavily impacted by avalanches in the past, including an event in 1988 that caused 42 deaths, people migrate to lower areas in February and March to stay safe from avalanches. In early spring water for irrigation here is still scarce, as glacier melt water peaks after May. By retaining, and in the process compressing, avalanche snow in narrow gullies where it generally descends, avalanche

snow subsequently melts much slower and can be tapped for irrigation water (Zakir, 2021; Figure 5). The technique of trapping avalanche ice-mass to create a more condensed solid water reservoir is known as avalanche harvesting, exploiting the reduced melt following from increased density and decreased surface area of the snow mass. It serves both risk reduction and produces melt water at the time of demand during the early months of spring for irrigation. An avalanche trapping net is knitted with high tensile gabion cables in gullies where there is hard bed rock on both sides. Each structure can have up to three to four

layers of netting, with a decreasing mesh size from up- to downstream. This is designed to reduce speed and pressure of the avalanche, and subsequently any downstream damage. Distances between layers vary between 20 to 50 meters. The trapping structure itself does not add additional risk in case it is destroyed during an avalanche event as it does not add any rolling mass. If the site for avalanche trapping is found and selected at an altitude of around 4000 m a.s.l., it can be expected that some of the condensed snow mass will remain and can grow with subsequent snowfall and avalanches, hence producing a permanent

mass that could be referred to as an *artificial glacier*, as in previous attempts using means to create such bodies of ice with water fountains (Balasubramanian et al., 2023).

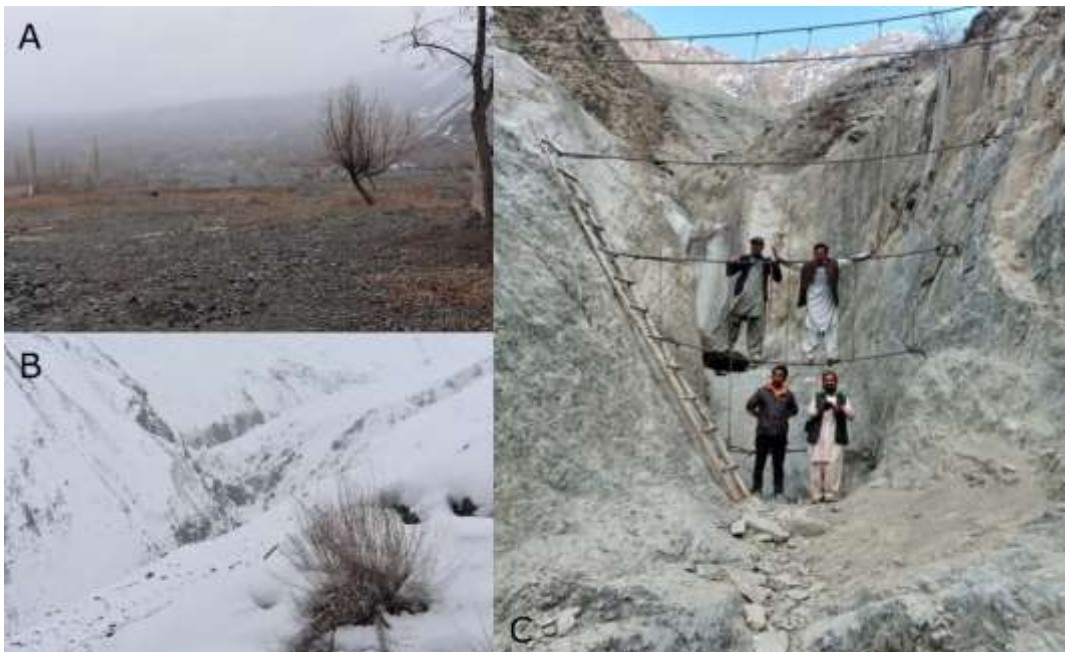

**Figure 5: (A) Lower agricultural areas repeatedly at risk from avalanches but also in need of irrigation water. (B) The gully in March. (C) One of the multiple nets installed along the gully to capture avalanche snow (bottom).**

## 7.3 Community based disaster management

Considering the increasing concern for snow hazards in Central Asia, FOCUS (now AKAH) has implemented a comprehensive risk anticipation and community preparedness program to reduce the adverse impact on vulnerable communities (Chabot and Kaba, 2016). Mountainous areas of Afghanistan, Pakistan, and Tajikistan have an avalanche readiness program through a Community-Based Disaster Management Program (CBDRM). FOCUS units have worked in identifying avalanche zones based on hazard, vulnerability and risk assessment (HVRA) maps with high frequency and high-intensity categorization. Similarly, awareness programs about avalanche-prone areas, dangers, early warning signs, and safe places/routes are being conducted. Moreover, the Community Emergency Response Team (CERTs) of each country are provided with proper training on early warning signs, basic snow rescue, and the use of basic measurement tools to quantify avalanche risk. It is also working on stockpiling safety equipment near avalanche-prone areas, archiving avalanche incidents to categorize them, assigning well-trained and dedicated staff for data management, gathering weather information, and reporting Emergency Response Officer. Moreover, FOCUS has also sourced an avalanche expert to help develop short and medium-term plans to mitigate the risk of avalanches in the highest-risk areas (Chabot and Kaba, 2016).

## 8 Discussion

### 8.1 Avalanche incidents and impacts

Our database provides a first comprehensive record of observed avalanches in HMA, including those that did not have any notable impact on livelihoods or infrastructure and information on impacts for those that did. Until the 2010s rarely more than 1 event was recorded in every two years in individual mountain ranges, which increased to more than 4 per year in the Pamir and more than 1 in the Hindukush (Figure 4), largely due to the commencement of a dedicated avalanche monitoring program. Records also increased elsewhere but it is impossible to say whether an increase in avalanche activity, an increased exposure or simply an increase in reporting (e.g. advent of social media resulting in more coverage also in local news) is responsible. Our records show that relying solely on peer reviewed publications would result in a considerable underestimation of events, capturing only 6% of all avalanches and 23% of all recorded fatalities. Relying on various news sources, which however often have conflicting documentation of what happened, requiring a lot of cross checkingcross-checking work, as well technical reports by agencies operating in hazard management in the region, proves essential.

Total fatalities in the region since the early 1970s end up at an average of 62 people killed each year, or 74 when we include deaths during mountaineering. Records in the Alps, an area considerably smaller and less populated than HMA show ~100 people who lost their lives between 1937 and 2015 (Techel et al., 2016). However, while in the Alps between 72% and 97% are related to mountaineering or recreation, this number is reverse in HMA (15%) and becomes even smaller (10%) when we consider that porters and guides died during work rather than recreation. In Switzerland, avalanche victims between 1946 and 2015 made up 37% of all natural hazard fatalities, more than any other. For HMA, not enough comprehensive data exist for

other hazards to make such a comparison. However data from GLOFs suggest a considerable lower number of deaths (907; excluding the Kedarnath event) for a much longer period (1833 – 2022) for the same ~~period~~ region (Steiner and Shrestha, 2022)~~(Shrestha et al., in review)~~. Landslides claim a lot more deaths in monsoon dominated India (~5000 between 1998 and 2018 excluding the Kedarnath event, Martha et al., 2021) and Nepal (2179 between 1978 and 2005; Petley et al., 2007).

Records for countries entirely in HMA suggest 67 deaths in Afghanistan, 50 in Bhutan, 809 in Nepal and 75 in Tajikistan between 2004 and 2010 (Petley, 2012). The relative lack of recreational sports in avalanche prone territory in the region hence makes avalanches relatively less deadly than in the Alps, also much less deadly than landslides in the region, but considerably more than GLOFs, a high mountain hazard that has received relatively more attention in literature (Steiner and Shrestha, 2022). Our data suggest that 56~~40~~ people died in avalanches while mountaineering in HMA, 191 (34%) of which were hired personnel

like guides and porters, emphasizing the danger of the job perceived as an important opportunity for some regions, especially in Nepal. An earlier study found slightly more fatalities for HMA (616), stemming from undocumented sources beyond peaks above 8000 m (McClung, 2016).

Previous work in North America (Schauer et al., 2021), Northern Europe (Hancock et al., 2021; Fitzharris and Bakkehøi, 1986) and the Alps (Jomelli et al., 2007) has shown the potential of investigating precursor meteorological data for avalanche

prediction using avalanche databases. In HMA such forecasting only exists in the Indian Western Himalaya as of yet. The present database would allow for a first analysis of such links. What is missing so far is an inventory of avalanches without significant recorded impacts for countries other than Afghanistan to make such an analysis robust. With such monitoring efforts being highly labour intensive and not possible for past events, the potential of remote sensing data would need to be further exploited, so far only done for limited areas in the region (Caiserman et al., 2022; Eckerstorfer and Grahn, 2021). Satellite

imagery has the capacity to scale up avalanche monitoring, especially thanks to the availability of numerous satellite datasets, both optical and radar. Due to the challenging nature of interpreting radar data for different types of avalanches (Eckerstorfer et al., 2016; Eckerstorfer and Grahn, 2021), databases relying on field observations will however also be necessary for validation purposes. Combining the available database of events with past regional climate data would allow to understand climatologies of avalanches in HMA better, an approach already employed in the Alps (Eckert et al., 2010). This could ~~would~~

allow for a better understanding of a link between a changing climate and the occurrence of avalanches, so far only investigated for a field site in India (Ballesteros-Cánovas et al., 2018) and one in China (Hao et al., 2023). More recently, distributed datasets on a number of variables including snow cover (Muhammad and Thapa, 2021), snow depth (Lievens et al., 2019), snow water equivalent (Smith and Bookhagen, 2018; Liu et al., 2021) and, locally, refreezing within the snowpack (Veldhuijsen et al., 2021) have become available. Such data could be combined with records of avalanche events to investigate

favourable conditions for avalanche occurrence, ideally in combination with more comprehensive assessments aided by remote sensing.

## 8.2 Mitigation, adaptation and scientific research

Mitigation and adaptation strategies in HMA have been highly diverse. No known efforts exist in Bhutan. In China a research station in the Tian Shan is tasked to investigate avalanche hazards specifically for roads (Hao et al., 2018). Any efforts and even accessible research in Central Asia, beyond the recent attention for glacier detachments (Leinss et al., 2021) is mainly limited to the work of FOCUS/AKAH spanning Afghanistan, Pakistan and Tajikistan (Chabot and Kaba, 2016; Bair et al., 2020). Recently, AKAH has installed two new stations in Gunt Valley, Tajikistan, where avalanches are frequent, in order to monitor the concurrent weather conditions. In Uzbekistan, various studies, employing RAMMS, have been conducted in order to improve the prediction of avalanche runout and eventually informing the construction of mitigation infrastructures. Long records of avalanches including observations, characterisation of morphometry, time of release as well as their weather conditions have been compiled (Gidrometeoizdat, 1984) and used for mitigation infrastructures construction, especially on Kamchik Pass in the Tian Shan (Semakova and Bühler, 2017; Semakova et al., 2018, 2009). In Western Himalaya and the Karakoram some preliminary studies for specific avalanche events employing RAMMS exist as well, however hampered in their quality by the lack of accurate snow depth data, general snow properties as well as local meteorological data (Gilany and Iqbal, 2019; Mahboob et al., 2015; Singh et al., 2020).

Afghanistan has by far the most comprehensive database on avalanches and records on their type and impacts. This reflects the country's dependence on snow as a resource as well as the considerable impact on livelihoods avalanches had in recent years (33% of all non-mountaineering related fatalities in HMA have been recorded in the country). The country now also has some record of avalanches retrieved from satellite imagery (Caiserman et al., 2022), which could provide a solution for regional approaches to map avalanche frequency for both the analysis of drivers as well as impacts. Considering that fatalities in Pakistan were relatively minor (366 deaths, 195 of which are related to military operations along the Line of Control), it is perhaps not surprising that the hazard has received relatively little attention in the country. However local adaptation attempts like avalanche harvesting are being pioneered here and show the role of Indigenous knowledge when it comes to adaptation to climate risks. This incongruency between local concern about avalanches as being a dominant hazard (Tuladhar et al., 2021) but a lack of any coordinated research or adaptation response coordinated by government bodies is also true for Nepal. Conversely, in India both adaptation and research measures have been top-down driven by the army and the government, providing a comprehensive approach to tackling the challenge of avalanche risks. This has so far not been expanded beyond the Western Himalaya, presumably also because in the East avalanches have been a minor concern so far. Contrary to Afghanistan no comprehensive database of avalanches without any notable impact exists in India, but the only study to date linking a changing climate to avalanche frequency has been conducted here (Ballesteros-Cánovas et al., 2018).

In HMA, Central Asia and other regions of the world, remote sensing is being used for avalanches mapping in three ways: (a) manual digitization of avalanches using high resolution from origin to runout zones (Hafner et al., 2021; Yariyan et al., 2020; Abermann et al., 2019), (b) automatic detection using radar images (Eckerstorfer et al., 2016; Yang et al., 2020; Martinez-Vazquez and Fortuny-Guasch, 2008; Vickers et al., 2016; Bühler et al., 2018; Karas et al., 2022) and (c), automatic detection

with optical images (Singh et al., 2019b; Caiserman et al., 2022). Some automatic hazard indication mapping frameworks developed for the European Alps (Bühler et al., 2018, 2022) could be potentially useful in deriving avalanche hazard maps in HMA, in areas where high-quality and high-resolution DEMs, snow data, and information on ground cover are available. These algorithms provide hazard maps for different return periods (Bühler et al., 2018) as well as considering the protective effect of forest cover (Bühler et al., 2022) . In the Western Himalaya, SASE has developed avalanche hazard data cards and digital avalanche atlases in a GIS environment for several road axes in Jammu and Kashmir and Himachal Pradesh which gives information on avalanche terrain and avalanche activities (DRDO, 2018). Another method by Barbolini et al. (2011) with two modules to perform avalanche hazard mapping over large undocumented areas is equally promising in mountainous areas. This method combines GIS tools, computational routines, and statistical analysis in order to provide a ''semi-automatic'' definition of areas potentially affected by avalanche release and motion and has proven to be effective in all cases where a preliminary cost-efficient analysis of the territories potentially affected by snow avalanche was needed and hence has great potential in areas like HMA. Several regional to local level avalanche hazard maps and susceptibility maps using frequency ratio, fuzzy and classical analytical hierarchy processes have already been generated in the Western Himalayas based on coarse to fine resolution digital elevation models for terrain parameters and other ground information (Kumar et al., 2017, 2016; Bühler et al., 2013; Kumar et al., 2018; Singh et al., 2019b; Snehmani et al., 2014) . Methodologies using airborne (Bühler et al., 2019; Lato et al., 2012; Korzeniowska et al., 2017) and spaceborne (Bühler et al., 2009) optical RS with very high resolution can detect avalanche deposits at large scale. These algorithms are promising in remote and inaccessible areas in HMA.

All approaches have their advantages and disadvantages. High resolution images enable a detailed outline of avalanches resulting in precise surface area estimation, but the financial cost of single images and limited spatial and temporal coverage as well as the time-consuming manual digitization limit their use on a regional scale. Radar sensors offer a global dataset of images which assess terrain evolution using backscatter changes between two sets of images. The change in snow surface area must be intense enough to be detected through radar images. Moreover, freely available radar products such as Sentinel-1 have commenced relatively recently (2014) and therefore do not allow for long-term monitoring of avalanches. On the contrary, optical images offer longer datasets, back to the 1980s. Automatic detection of avalanches using optical images can however be difficult as the contrast between snow avalanches zones and normal snow cover is often difficult to discern. It hence remains up to stakeholders to choose which products and methods match their needs, objectives and most importantly, human and financial resources. We suggest however, that based on studies in the region as well as in the European Alps, sufficient openly accessible guidelines as well as data are available to allow policy makers and agencies responsible for disaster risk reduction (DRR) in the region to develop action plans to mitigate avalanche hazards.

Considering that all aspects of avalanche risk management have been executed somewhere in the region, ranging from local practices on adaptation to state of the artstate-of-the-art analysis of satellite imagery, but nowhere all of them are exploited in tandem, it seems prudent to learn from experiences across borders to further advance our understanding of avalanche dynamics in the region. While the runouts of avalanches are rarely of transboundary nature and potentially will only ever be for large scale ice avalanches or glacier detachments that are able to mobilise a lot of solid material in the downstream, seasons with

heavy avalanche activity generally extend across borders (e.g. Nepal and India, Pakistan, Tajikistan and Afghanistan) and
ramifications in risk (e.g. effect on international tourism in the Himalaya, labour migration in the Hindukush) are of
transboundary nature.

**9 Conclusion**

In this study we for the first time investigate all accessible snow and ice avalanche records from the HMA region as well as
previous scientific research, local knowledge and existing or previously active adaptation and mitigation solutions.
Our database spansning half a century between 1972 and 2022 and records 681 individual events (88% snow, 0.5% ice
avalanches; excluding mountaineering accidents on high peaks), of which 21% had some recorded impact and in 17% fatalities
were recorded. More than 3100 people were killed by avalanches, more than 62 in each year on average, most of them in
Afghanistan (1057; 34%) and India (952; 30%). This suggests avalanches to be an especially deadly hazard in high mountain
environments in the region, causing considerably more fatalities than, for example, glacier lake outburst floods. On average,
each avalanche in our database killed 27 peopleAn average of 27 people died in each individual avalanche, making them also
especially high impact events. Nearly 1900Approximately 1800 livestock killed in 24 events are also noteworthy, as this impact
to livelihoods has been mentioned as an especially problematic aspect of the hazard in the region.
More than 500 people have been killed in avalanches while climbing mountain peaks above 4500 m a.s.l., a third of which
were porters or guides employed during these endeavours.
Scientific studies on avalanches with field investigations are so far limited to research stations on the Tibetan Plateau as well
as the Indian Western Himalaya. As a result, modelling studies that have been carried out in a number of locations remain
relatively rudimentary as information on snowpack and local climate is lacking.
Studies investigating the role of avalanches for glacier mass balances are still limited but suggest large differences in their
relative contribution between the monsoon dominated Himalaya, where also steep and very high headwalls surrounding glacier
tongues are likely more common, conducive to avalanche occurrence (>75%) and the continental and less topographically
extreme Tiaen Shan (<15%). Future studies should investigate their relevance on a more regional scale, also considering the
increasing dynamic disconnect of accumulation and ablation areas in many glaciers in the region.
Conversely a number of adaptation as well as mitigation measures have already been taken up. In India and China this has
received large scale funding for specific regions where avalanches have been identified as potential threats for road
infrastructure and includes dedicated monitoring programmes in coordination with scientists as well as investments into state
of the artstate-of-the-art structural measures. Non-governmental efforts have however been equally successful. The monitoring
of avalanches in Afghanistan is unparalleled in the region and carried by local stakeholders. It is the only country that records
individual avalanche events irrespective of impact, an effort that is incredibly valuable for future studies investigating
meteorological drivers. Other examples of locally carried initiatives like the harvesting of avalanche deposit for meltwater to

be used in high mountain irrigation in Pakistan are also a testament for the existence of local solutions in addressing the hazard that should be investigated for potential scaling to other localities in future.

The database of avalanche events, including geolocation, timing, type of avalanche and recorded impacts allows for future investigations of climatologies of avalanche events in HMA. As a first step they can be used to validate automated mapping efforts using remote sensing on a regional scale and subsequently correlated with distributed climate and snow datasets to

understand important regional drivers for avalanche occurrence. This would provide a basis for more robust forecasting of avalanche danger in different parts of HMA, where avalanche risks have been identified as a concern. Efforts by national agencies tasked with hazard monitoring should further invest in pairing scientific investigations of avalanche hazards with developing adaptation and mitigation solutions, an approach that has seen success in other mountain regions. Based on the impacts we record here avalanches had in the past in HMA, we suggest that the hazard should be given a higher priority in

DRR strategies of the different governmental and non-governmental stakeholders in the domain of high mountain hazards. Methodologies as well as data are available to address the issue with both remote sensing data as well as field-based installations. National and local solutions for such avalanche warning and forecasting should always keep in mind already existing local adaptation solutions and knowledge with respect to the hazard. Future interdisciplinary studies should further investigate local solutions in response to avalanche hazards as well as record experiences with the hazard and whether and

how they have changed in the more recent past. This would allow to understand whether avalanche risks are changing (either due to climate or more socio-economic exposure) and help in building future adaptation solutions.

## 10 Data Availability

The data collected for this review is available at DOI: 10.5281/zenodo.7066941 and https://github.com/fidelsteiner/HiAVAL (Steiner and Acharya, 2022), where it will be updated regularly. The data can also be accessed interactively on

https://experience.arcgis.com/experience/cf2ac60cac6c4a94b4a1cdf09d74aba5

## Acknowledgements

AA compiled the review, analysed the data and wrote the manuscript. JFS supported in data compilation and analysis and co-wrote the manuscript. KMW provided avalanche records from Afghanistan and all authors contributed to writing the manuscript. The authors thank Inka Koch and Denis Samyn for earlier contributions to this work. The views and interpretations

in this publication are those of the authors and are not necessarily attributable to ICIMOD. We are grateful to the reviews of Yves Bühler and Markus Eckerstorfer and additional comments from Wolfgang Schwanghart, which helped us to considerably improve the manuscript.

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
