# Peer review of "Review article: Snow and ice avalanches in high mountain Asia – scientific, local and indigenous knowledge"

_Natural Hazards and Earth System Sciences, 2022_

## Author Comment (AC1)

Dear Authors!

I am very impressed by the amount of data you have gathered on avalanche events in HMA. Your GIS based database is great and a good starting point for further studies and initiatives as you write in your manuscript. This work is of high value and very timely for the people living in these areas and their livelihood being threatened by snow and ice avalanches (and other slope and glacier hazards).

I suggest minor revision that mostly concerns some clarifications, writing (I am not an English native speaker myself) and the use of technical terms.

Dear Markus,

we are very grateful for the careful reading of the manuscript, especially pointing out numerous incidences where different formulations from publications we could not bring to a more standardized nomenclature. We address each individual issue raised below, with your comments followed directly by our response in red.

Doesn't – "does not" in several places

Thanks, we have now checked throughout and amended this. We think that actually it should be 'does not' for formal standards and we have changed the 'doesn't' to that effect. We hope that follows the NHESS standard.

40: consider splitting up this very long sentence

Thanks, we have split this up into two, subsequently shifting the citation.

43: since you mention avalanches for the first time here, maybe write "snow avalanches (hereafter also called avalanches)"

Thanks, amended.

46: I would delete Eckerstorfer and Malnes, 2015 here, since it is not the correct paper to cite on how avalanches affect socio-economic activities. McClung's Handbook is fine here.

Thanks for the suggestion, we have removed it here.

55: not sure how important these past death tolls are here. Consider looking up major avalanche accidents in Asia, for example in Panjshir, Afghanistan in 2015.

We understand that these are numbers not relevant to Asia here, but we feel they make sense for an agenda setting, as indeed in Asia this issue of soldiers caught in avalanches has repeated itself. Therefore, we would prefer keeping it. We have discussed all other major avalanches in the HMA in the later sections and referring to them here would jump ahead.

78: I do not think that avalanche related climate change studies are that limited anymore. There are numerous studies from Switzerland and France that should certainly be mentioned here.

We have done a careful search on the avalanche-climate change nexus again and there indeed remains relatively limited literature on the topic. We have now added (Giacona et al. 2021; Eckert et al. 2013) which we previously missed and made the addition that research on the topic largely is focused on the French Alps. We also added the very recent (Hao et al. 2023) under results since this is for the region, and add the relevant discussion. There is one more study (Strapazzon et al. 2021), however it deals with the impact on accident severity rather than frequency or nature of avalanches and we hence do not refer to it here.

108: until here I thought your review would only concern ice avalanches. I believe you should consider giving an overview over ice avalanches in the introduction, similar to the comprehensive introduction to snow avalanches.

Thanks for pointing this out, this should have indeed been made clearer. We have now added a separate line to the Introduction where we mention the separate line of research for ice avalanches (L79 – L86). This is considerably shorter than the general introduction as research on ice avalanches remains more limited, but of course is of crucial importance. We circle back to this issue in Section 2 from L117.

147: depth-hoar avalanches is not a commonly used term. Could you write "avalanches that release on the ground" or "full depth avalanches".

Thanks, we have amended this.

Section 3.1 you are switching a bit between past and present tense. Consider writing it all in past tense.

Thanks for noticing, we have now throughout put this into past tense except for generally applicable statements that are kept in present tense.

151: it is not correct that a deep snowpack favors depth hoar growth. In fact, it is the opposite. A thin snowpack allows for a large temperature gradient between ground and atmosphere and thus kinetic growth of snow crystals into depth hoar.

Yes, it is true, here it has been written awkwardly, the author mentions about the condition favoring for depth hoar avalanche (full depth). Now, it has been simplified

156: it does not make sense that SDAs (surface layer dry slabs) occurred with a persistent weak layer. However, it makes sense that they were triggered by heavy snowfall and wind. Consider deleting the information on persistent weak layers. These would be present with FDA for example.

Thanks for noticing, this may have been inconsistent in the original paper and we have amended accordingly.

159: if a storm triggered 37 avalanches, it does not make sense that half of them were triggered by precipitation. What about the other half?

Thanks for noticing, this was indeed formulated awkwardly. The study found that general threshold during any storm needs to be present (high intensity precipitation) but identified precipitation as the final trigger in 18 and high temperatures in 10 cases, 6 were triggered by earthquakes and 3 events by other triggers. We have adapted the text accordingly.

162: sentence is not understandable. Please rewrite.

Sentence revised.

167: please check again if these avalanches predominantly released between 25 and 35 degrees. Where these wet snow avalanches or slush flows?

The authors do refer to wet avalanches here, not slush flows and from the study site we also believe that the latter wouldn't be possible. The study area has two classes with high percentages/5 classes of avalanche release slopes in descending images of two study sites; Kizilkeya and Aktep in West Tian Shan Mountains.

Kizilkeya:

      35-50: 38%

      25-35:54%

Aktep:

    35-54: 36%

    25-35: 53%

172: consider splitting up this very long sentence

We split the sentence into two.

176: not really sure what a moist slab avalanche is. In my opinion the classification is either wet or dry.

We agree that the 'moist' classification used here may not be significant or entirely useful and we have removed it in this context.

178: we usually talk about wind-loaded slopes. Thaw avalanches is not a term commonly used. I guess you mean wet snow avalanches?

We have used terms used in the publications but agree that it would be helpful to converge on standardized terms here. We have adapted this accordingly.

189: what do you mean by weakened layers?

We here refer to the snowpack being weakened as water content rises and have clarified in the text.

195: consider splitting up this very long sentence

We have split the sentence in two.

200: contradicting who? Your opinion?

The contradiction here would be to communities' expectations as well as the simple notion that one would expect less avalanche activity with less snow. However we agree that here this can not be juxtaposed against some published statement and we hence remove it.

Figure 1a: this seems to be a glide crack that precauses a glide avalanche.

Thanks, we have updated the caption.

Figure 1d: I wonder if the avalanche eroded the gully. Would you rather not think the gully was already there, caused by some other process?

This is a god point and we didn't mean to suggest that a single avalanche caused the gully. Having observed the particular location over many seasons we however know that it was gradually formed by repeat avalanches as well as the subsequent debris flows caused by excessive melt water. We have now made that clear in the caption.

Figure 3a: the color indicating glaciated areas is not the same as in the legend.

Not visible at this scale, but we have now adapted the label also simply to blue.

Figure 3b: how do I interpret the map? If a circle over an area shows 30-40 fatalities, are all of these fatalities from a single event, or the sum of fatalities from multiple events over multiple years?

The circle sizes are for individual events. We have now clarified that in the caption.

Figures 3 and 4: what do you mean by "some recorded impact"?

This refers to either injured, killed people, killed livestock or damaged or destroyed infrastructure. We have clarified that in the caption.

Figure 4b: what are impacts by month?

This is the same data as in 4a but aggregated by recorded month to show the seasonality of avalanche accidents in the region. We have added 'recorded' to go in line with the added explanation in 4a.

Figure 4a: the figure is not easy to read and interpret. I am not sure there is much information to be gained from Figure 4b. maybe consider showing the stacked fatalities per year in Figure 4b instead.

We have now shown the stacked fatalities per year

Table 2: if you do not know the type of slope hazard (NA), why are these events included then in the table? Does it make sense to mention "slab avalanche" in the table? I guess its falls into the snow avalanche category anyway.

All events with NA are still avalanches (i.e. either snow or ice), the records however do not specify which. In some cases (e.g. Salang in Afghanistan) we can be sure it must be a

snow avalanche as no ice is close but we stick here to the records given as in others (e.g. Lahaul, Gokyo) it could be both or a mix of both.

For 'slab avalanches' we simply had this added information available, but to not confuse the reader here we hv changed this to 'snow avalanche'. The detail is retained in the HiAVAL database.

330: you initially stated that you are not considering high altitude mountaineering avalanche fatalities. However, here you have a whole section on it. I find it interesting and I think you should slightly rewrite your scope of the study in the introduction.

We have indeed recorded all mountaineering fatalities as well (see L111f). When talking about impacts and e.g. climate change or seasonality we believe that clustering them under the total number is not helpful as it would skew conclusions e.g. for adaptation by country towards the issue. Most of these fatalities happen in Nepal, a country that otherwise has fewer avalanche casualties than India or Afghanistan, which leads to the conclusion that avalanche deaths here are rather driven by willful exposure rather than the presence of avalanches per se. All data is also recorded in the database and both numbers are also stated in the abstract to make the total clear.

Section 6: interesting topic, however, very thinly described. Could you add some references here for further reading?

Information is limited in this regard; however, we have now added additional literature following your recommendation. It remains difficult to come to actual conclusions in this regard as attempts have been scattered.

453: love the story about people basically growing rock glaciers. Have heard that in some lectures before!

Thank you.

---

## Author Comment (AC2)

Dear Yves,

We are very grateful for your careful read of the manuscript and acknowledging the contribution to an understudied part of the cryosphere in HMA. We have considered all the suggestions made and respond to them individually below, with our response marked in red.

The paper entitled «Snow and ice avalanches in high mountain Asia – scientific, local and indigenous knowledge» reviews the known avalanche events in high mountain Asia, develops a database and discusses mitigation strategies. This is a very meaningful and important contribution to a topic not yet well investigated in this region. Therefore, I would recommend this paper for publication after adding some more important sources and improving some figures as well as an extended discussion on avalanche hazard indication mapping. Her are my specific inputs:

Abstract:

L 24: you mean more densely populated right? I think the European Alps are by far more populated than HMA. You write "in the less populated European Alps" is this a mistake?

Delineations of mountain areas and hence their associated populations are difficult and there exists no one consensus (Thornton et al. 2022). But considering that Nepal alone, which in many definitions falls completely within HMA) has 30 million inhabitants, Tibet more than 3 million, Gilgit Baltistan 1.5 million, Tajikistan ca 10 million, Kabul at 1800 m a.s.l. 4 million etc. we think HMA has indeed more people than the Alps, which in the delineation of the Alpine Convention stands at 14 million[1]. In both cases these areas include land that is relatively low (Kathmandu, Luzern etc) but fall within the global delineations. We are not so sure if can make a blanket statement on density as that would vary within the regions considerably.

Introduction:

L75: Here I would propose to add some more important references on avalanche mapping by remote sensing as they bare a big potential for future applications in HMA: (Lato et al., 2012; Eckerstorfer et al., 2019; Bühler et al., 2009; Korzeniowska et al., 2017; Bühler et al., 2019)

Thanks for these added suggestions. We have included this and refer to it further in the Discussion.
* * *
[1] https://www.alpconv.org/en/home/topics/population-and-culture/

Recorded                    snow                    and                    ice                    avalanches:

L250: Fig 3 is not well readable, in particular the legends must be larger.

L255 Fig 4 is also only poorly readable, the fonts must be bigger.

Thank you, this has been adapted.

Discussion:

Here I miss a discussion on hazard maps and hazard indication maps. For he Alps, approaches have been developed to automatically generate hazard indication maps based on digital elevation models (Bühler et al., 2022; Bühler et al., 2018). Such approaches would be very helpful for HMA regions. Also, more simple approaches such as slope angel maps would be helpful for the planning of mitigation measures and could have a high impact. This should be discussed in more detail. Are there existing hazard indication maps for HMA? As far as I know the Aga Khan foundation is also doing hazard indication mapping for avalanches for example in Afghanistan.

Thank you, we now expanded the discussion here considerably, would however like to note that it is simply also a lack of responses to avalanche hazards in the region that makes this difficult to fill. Future work on proposed ways forward for the region would be definitely welcome and we have noted that in the Conclusion as well.

References

Bühler, Y., Hafner, E. D., Zweifel, B., Zesiger, M., and Heisig, H.: Where are the avalanches? Rapid SPOT6 satellite data acquisition to map an extreme avalanche period over the Swiss Alps, The Cryosphere, 13, 3225-3238, 10.5194/tc-13-3225-2019, 2019.

Bühler, Y., Hüni, A., Christen, M., Meister, R., and Kellenberger, T.: Automated detection and mapping of avalanche deposits using airborne optical remote sensing data, Cold Regions Science and Technology, 57, 99-106, 10.1016/j.coldregions.2009.02.007, 2009.

Bühler, Y., von Rickenbach, D., Stoffel, A., Margreth, S., Stoffel, L., and Christen, M.: Automated snow avalanche release area delineation – validation of existing algorithms and proposition of a new object-based approach for large-scale hazard indication mapping, Natural Hazards and Earth System Sciences, 18, 3235-3251, 10.5194/nhess-18-3235-2018, 2018.

Bühler, Y., Bebi, P., Christen, M., Margreth, S., Stoffel, L., Stoffel, A., Marty, C., Schmucki, G., Caviezel, A., Kühne, R., Wohlwend, S., and Bartelt, P.: Automated avalanche hazard

indication mapping on a statewide scale, Nat. Hazards Earth Syst. Sci., 22, 1825-1843, 10.5194/nhess-22-1825-2022, 2022.

Eckerstorfer, M., Vickers, H., Malnes, E., and Grahn, J.: Near-Real Time Automatic Snow Avalanche Activity Monitoring System Using Sentinel-1 SAR Data in Norway, Remote Sensing, 11, 10.3390/rs11232863, 2019.

Korzeniowska, K., Bühler, Y., Marty, M., and Korup, O.: Regional snow-avalanche detection using object-based image analysis of near-infrared aerial imagery, Nat. Hazards Earth Syst. Sci., 17, 1823-1836, 10.5194/nhess-17-1823-2017, 2017.

Lato, M. J., Frauenfelder, R., and Bühler, Y.: Automated detection of snow avalanche deposits: segmentation and classification of optical remote sensing imagery, Natural Hazards and Earth System Sciences, 12, 2893-2906, 10.5194/nhess-12-2893-2012, 2012.

---

## Author Response (AR2)

Dear Editor,

we are grateful for the additional close reading and would like to document our changes below.

Following the request from Dr. Ali and Dr. Hussain, their respective order in the author list has been switched. They have both contributed equally, however Dr. Ali is up for tenure and this makes a difference in evaluation. We hope this is understandable.

We have also checked through the complete manuscript for minor edits (especially missing commas etc).

Line 15: Avoid the twice use of poorly in this sentence.

Thank you, amended accordingly

Line 63: remove comma after reference

Thank you, this has been removed

Line 65: The abbreviation NN is introduced here, but rarely (actually only in one paragraph starting in 385) used. Consider avoiding it because readers probably won't remember when reaching section 6.
Thank you, amended accordingly

Line 104: variety of different sources. Remove either variety or different.

Thank you, this has been adapted

Line 417: AWS - this abbreviation has not been introduced before. But there's no need becaused it's only used once. Hence, write "automated weather stations".

Thank you, amended accordingly

Line 656: Our database spans half a century between 1972 and 2022 and records ...

Thank you, amended accordingly

660: On average, each avalanche in our database killed 27 people... Note that otherwise, the sentence can be misinterpreted that each avalanche kills on average 27 people.

Thank you, this has been adapted

Fig. 3: The figure is quite difficult to read. Perhaps you can increase the readability by using an image in the background with less saturation. Since many of the circles and triangles overlap, it may also be advisable to use some transparency of the markers. This enables readers to appreciate how much these markers overlap. The last sentence in the caption could

be rephrased: "The black outline shows regional outlines of High Mountain Asia, with numbers referring to region-IDs (Bolch et al., 2019) used in the avalanche database."

The figure has now been updated as suggested, and the caption has been clarified.

Fig. 4: You may want use the start and end date of each decade as x-labels (e.g, 1970-79, 1980-89, ...) in panel B. This will also show that the last value for 2020 is aggregated only over a few years compared to the other bars. In C, I think that the relative number of events (Nmonth/Nyear) would provide better insights into the possibly variable seasonal frequency distribution in each region.

We have added the decadal ranges in panel B. We have also tried to standardize the seasonal occurrence by the number of years they have occurred in as we agree that in principle this would be a fairer comparison (Figure 1). However as is obvious this brings the 'mean' occurrence to 1 for all months and regions after April, as those months (per region) simply only had one event each. To underline this point (that avalanche seasonality beyond January to April can not be deduced due to the limited number of events) we have however added a line explaining at L275. In the process we also realized that Tian Shan was not consistently spelled, now amended.

[Figure]

*Figure 1: Seasonal avalanches per region, divided by the number of years any monthly avalanche has occurred.*

In Line 646, some formatting error occurred. Please correct before resubmission.

Thank you, this has been adapted.

There has already been quite some interest for this publication in the last weeks and during one interview with a journalist who was completely focusing on climbing, we realized that during writeup of the manuscript a reference to a previous paper that discussed deaths in avalanches in climbing only had completely slipped (McClung 2016). As we already explained to one of the reviewers, it was important for us that this manuscript looks at avalanches primarily and does not overtly focus on climbing deaths, as we feel the earlier aspect had been completely missing. Hence there is no added discussion we have on the matter apart from the fact that the previous paper found a similar fraction of hired personnel affected and it also does not change anything in regard to our review of the issue. We have now added this in L363 and L571 again in the Discussion, and hope that this late addition will be acceptable. As you will see the numbers cited there are slightly higher than our record, however, as the documented cases where sources are given in (McClung 2016) match exactly with our records (not surprising as we rely on the same sources) we are unable to say where the added numbers come from, maybe emphasizing again the need for open data.

Kind Regards,

Anushilan Acharya

On behalf of all co-authors

References

McClung, D. M. 2016. "Avalanche Character and Fatalities in the High Mountains of Asia." *Annals of Glaciology* 57 (71): 114–18. https://doi.org/10.3189/2016AoG71A075.